# CAE: Repurposing the Critic as an Explorer in Deep Reinforcement Learning

**Yexin Li**  *liyexin@bigai.ai*
*State Key Laboratory of General Artificial Intelligence, BIGAI*

**Reviewed on OpenReview:** *https://openreview.net/forum?id=54MODO2xC2*

## Abstract

Exploration remains a fundamental challenge in reinforcement learning, as many existing methods either lack theoretical guarantees or fall short in practical effectiveness. In this paper, we propose CAE, *i.e.*, the Critic as an Explorer, a lightweight approach that repurposes the value networks in standard deep RL algorithms to drive exploration, without introducing additional parameters. CAE leverages multi-armed bandit techniques combined with a tailored scaling strategy, enabling efficient exploration with provable sub-linear regret bounds and strong empirical stability. Remarkably, it is simple to implement, requiring only about 10 lines of code. For complex tasks where learning reliable value networks is difficult, we introduce CAE+, an extension of CAE that incorporates an auxiliary network. CAE+ increases the parameter count by less than 1% while preserving implementation simplicity, adding roughly 10 additional lines of code. Extensive experiments on MuJoCo, MiniHack, and Habitat validate the effectiveness of CAE and CAE+, highlighting their ability to unify theoretical rigor with practical efficiency.

## 1 Introduction

Exploration in reinforcement learning (RL) remains a fundamental challenge, particularly in sparse-reward environments. Although algorithms such as DQN (Mnih et al., 2015), PPO (Schulman et al., 2017), SAC (Haarnoja et al., 2018), DDPG (Lillicrap et al., 2016), TD3 (Fujimoto et al., 2018), IMPALA (Espeholt et al., 2018), and DSAC (Duan et al., 2021; 2023) have achieved impressive performance on tasks like Atari games (Mnih et al., 2013; 2015), StarCraft (Vinyals et al., 2019), Go (Silver et al., 2017), *etc.*, they often depend on rudimentary exploration strategies. Common approaches, such as $\epsilon$-greedy or injecting noise into actions, are often sample-inefficient and struggle in environments with delayed or sparse rewards.

For decades, exploration strategies with provable optimality guarantees have been well established in tabular RL settings (Kearns & Singh, 2002). More recently, methods with provable regret bounds have been progressively extended to RL with function approximation, including linear functions (Osband et al., 2016; 2019; Jin et al., 2018; 2020; Kamyar & Animashree, 2018; Agarwal et al., 2020), kernel-based models (Yang et al., 2020), and neural networks (Yang et al., 2020). While linear and kernel-based approaches make strong assumptions about the RL functions, the provable method based on neural networks suffers from prohibitive computational costs, specifically $O(n^3)$, where $n$ is the number of parameters in the network. Subsequently, some studies (Ash et al., 2022; Ishfaq et al., 2024a) propose algorithms with theoretical optimality guarantees under the linearity assumption and extend them directly to deep RL without further proof. Other works (Ishfaq et al., 2021; 2024b) derive provable regret bounds for their deep RL methods, but these approaches are either practically burdensome or rely on unknown sampling errors.

More practical exploration approaches typically rely on heuristic mechanisms, giving rise to a variety of empirically successful methods, including Pseudocount (Bellemare et al., 2016), ICM (Pathak et al., 2017), RND (Burda et al., 2019b), RIDE (Raileanu & Rocktäschel, 2020), NovelD (Zhang et al., 2021a), AGAC (Flet-Berliac et al., 2021), and E3B (Henaff et al., 2022; 2023). These methods introduce intrinsic

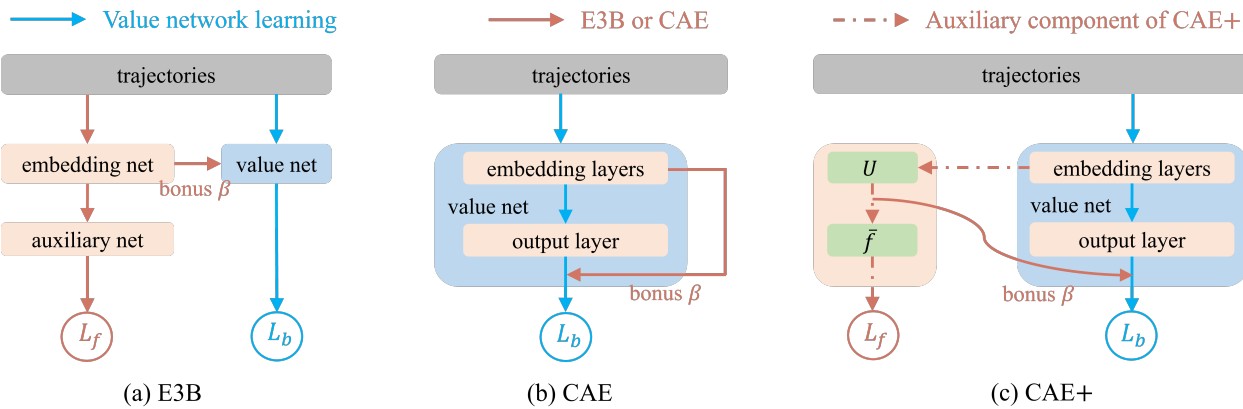

Figure 1: Comparison of baselines, such as E3B, with CAE and CAE+. $L_b$ denotes the Bellman loss used to update the value function, while $L_f$ refers to the loss of the auxiliary network. Unlike E3B, which requires a separate network to compute exploration bonuses, CAE exploits the embedding layers of the value network for this purpose. CAE+ further enhances CAE by incorporating a compact, lightweight auxiliary network, $f = \bar{f} \circ U$, improving performance in sparse-reward environments with only a minor increase in parameters.

bonuses to encourage agents to visit novel states. For instance, RND computes the exploration bonus using the prediction error of a randomly initialized target network. Despite their empirical effectiveness, such approaches often distort the extrinsic reward signal and generally lack formal theoretical guarantees. In contrast, potential-based reward shaping methods, such as Liberty (Yiming et al., 2023) and EME (Yiming et al., 2024), provide stronger theoretical justification but are often challenging to implement in practice. Moreover, all of the above methods require training auxiliary networks beyond the standard policy and value networks used in deep RL, resulting in substantially increased computational overhead.

In this work, we aim to combine the strengths of both theoretically grounded and empirically effective exploration methods. Provably efficient approaches are fundamentally rooted in the theory of Multi-Armed Bandits (MAB) (Li et al., 2010; Chu et al., 2011; Agrawal & Goyal, 2013; Wen et al., 2015; Zhang et al., 2021b; Zhou et al., 2020). Building on this foundation, we hypothesize that advanced techniques from **neural MAB** can be effectively adapted for exploration in **deep RL**. Recent studies (Zahavy & Mannor, 2019; Riquelme et al., 2018; Xu et al., 2022) suggest that decoupling deep representation learning from exploration strategies holds promise for achieving efficient exploration in neural MAB settings.

Motivated by these insights, we propose CAE. Unlike existing methods that train additional embedding networks to generate exploration bonuses, CAE leverages the embedding layers of the value networks in the RL algorithms and employs MAB techniques to produce exploration bonuses. To ensure the practical stability of CAE, we adopt an appropriate scaling strategy (Welford, 1962; Elsayed et al., 2024) to process the bonuses. Consequently, CAE introduces no additional parameters beyond those in the original algorithms, showcasing that RL algorithms inherently possess strong exploration capabilities if their learned networks are effectively leveraged. Moreover, CAE is simple to implement, requiring only about 10 lines of code. A comparison between CAE and existing methods is in Figure 1.

For tasks with complex dynamics and very sparse rewards, learning value networks is challenging, impeding exploration based on them. Accordingly, we propose an extended version CAE+, as illustrated in Figure 1. CAE+ integrates a compact, lightweight auxiliary network to facilitate the learning process. The structure of the auxiliary network is carefully designed to prevent severe coupling between the environment dynamics and the returns, thereby further enhancing the performance of CAE+. Remarkably, this addition increases the parameter count by less than 1% and requires only about 10 extra lines of code, thus preserving the simplicity and lightweight nature.

Our experiments cover a diverse set of benchmarks, *i.e.*, MuJoCo, MiniHack, and Habitat, representing dense-reward, sparse-reward, and reward-free environments. CAE improves the performance of state-of-the-art RL baselines, including PPO (Schulman et al., 2017), SAC (Haarnoja et al., 2018), TD3 (Fujimoto et al., 2018),

Table 1: Method comparison. **Linear function** indicates whether the theoretical guarantees apply to RL algorithms with linear function approximation. **Network-based function** specifies whether the guarantees extend to RL algorithms using network-based function approximation. **Toy task** denotes whether the method can be effectively applied to simple deep RL tasks. **Practical complex task** indicates whether the method remains practically feasible when deployed in realistic, large-scale deep RL environments.

| Method | Provable | | Empirical with Deep RL | |
|---|---|---|---|---|
| | Linear function | Network-based function | Toy task | Practical complex task |
| LSVI-UCB (Jin et al., 2020) | ✓ | ✗ | ✗ | ✗ |
| NN-UCB (Yang et al., 2020) | ✓ | ✓ | ✗ | ✗ |
| OPT-RLSVI (Andrea et al., 2020) | ✓ | ✗ | ✗ | ✗ |
| LSVI-PHE (Ishfaq et al., 2021) | ✓ | ✓ | ✓ | ✗ |
| BDQN (Kamyar & Animashree, 2018) | ✓ | ✗ | ✓ | ✓ |
| ACB (Ash et al., 2022) | ✓ | ✗ | ✓ | ✓ |
| LMCDQN (Ishfaq et al., 2024a) | ✓ | ✗ | ✓ | ✓ |
| CAE | ✓ | ✓ | ✓ | ✓ |
| CAE+ | ✓ | ✓ | ✓ | ✓ |

and DSAC (Duan et al., 2021; 2023). Additionally, CAE+ demonstrates robust performance, consistently outperforming E3B Henaff et al. (2022; 2023), the state-of-the-art exploration method for MiniHack and Habitat, across all evaluated tasks. These results highlight the superior reliability and effectiveness of CAE and CAE+ in diverse RL scenarios.

In summary, we make three key contributions. First, we propose lightweight CAE and CAE+, which enable the use of linear MAB techniques for exploration in deep RL. By adopting a scaling strategy and carefully designing the small auxiliary network for CAE+, we ensure both practical stability and functionality in environments with dense and sparse rewards. Second, our theoretical analysis demonstrates that any deep RL algorithm with CAE or CAE+ achieves a sub-linear regret bound over episodes. Finally, experiments on MuJoCo, MiniHack, and Habitat validate the effectiveness of CAE and CAE+, showcasing their superior performance. Our code is available at https://github.com/liyexn/Critic-as-an-Explorer.

## 2 Related Work

**Multi-armed bandits.** MAB algorithms address the exploration-exploitation dilemma by making sequential decisions under uncertainty. LinUCB (Li et al., 2010) assumes a linear relationship between arm contexts and rewards, enabling efficient exploration via uncertainty quantification in the estimated parameter space and ensuring a sub-linear regret bound (Chu et al., 2011). To relax the linearity assumption, KernelUCB (Valko et al., 2013; Chowdhury & Gopalan, 2017) and NegUCB (Li et al., 2024) transform contexts into high-dimensional spaces and apply LinUCB to the mapped contexts. Neural-UCB (Zhou et al., 2020) and Neural-TS (Zhang et al., 2021b) leverage neural networks to model the complex relationships between contexts and rewards. However, their computational complexity of $O(n^3)$, where $n$ denotes the number of network parameters, limits their scalability in real-world applications. Neural-LinTS (Riquelme et al., 2018) and Neural-LinUCB (Xu et al., 2022) mitigate this limitation by decoupling representation learning from exploration strategies, improving the practicality of neural MAB.

**Provable exploration in RL.** Provably efficient exploration methods (Kearns & Singh, 2002; Osband et al., 2016; 2019; Jin et al., 2018; 2020; Agarwal et al., 2020; Cai et al., 2020; Daniil et al., 2022; 2023) often face empirical limitations or remain primarily theoretical, with limited applicability to deep RL. Some studies (Alessio & Filippo, 2024; Kamyar & Animashree, 2018; Ash et al., 2022; Ishfaq et al., 2024a) propose methods with theoretical guarantees under tabular or linearity assumptions and extend them directly to deep RL settings. Other works (Ishfaq et al., 2021; 2024b) provide provable bounds for deep RL, but these approaches may suffer from any of the following limitations: computationally prohibitive, practically burdensome, or reliance on unknown sampling errors. A comparative analysis is presented in Table 1.

**Practical exploration in deep RL.** Empirically successful methods (Bellemare et al., 2016; Pathak et al., 2017; Burda et al., 2019a; Raileanu & Rocktäschel, 2020; Burda et al., 2019b; Zhang et al., 2021a; Flet-Berliac et al., 2021; Henaff et al., 2022; 2023; Jarrett et al., 2023; Yuan et al., 2023) typically generate exploration bonuses to encourage agents to visit novel states. However, these approaches often lack rigorous theoretical foundations. In contrast, methods inspired by potential-based reward shaping (Andrew et al., 1999), such as Liberty (Yiming et al., 2023) and EME (Yiming et al., 2024), offer stronger theoretical grounding, but are challenging to implement in practice. Moreover, both classes of methods generally require training a substantial number of additional parameters. In comparison, CAE and CAE+ utilize MAB techniques, assisted by embedding layers within the RL value networks, providing empirical benefits with minimal additional parameters. Figure 1 illustrates the differences among various exploration methods, while Table 2 summarizes their additional networks and parameters.

Table 2: Comparison of exploration methods on MiniHack. **Networks**: additional networks beyond those in the base RL algorithm IMPALA (Espeholt et al., 2018), which contains $25, 466, 652$ parameters; **# Params**: the number of additional parameters introduced by the exploration method. Networks in **bold** represent those with significant parameters, while those in gray indicate substantially fewer parameters.

| Method | Networks | # Params | Params ↑ |
|---|---|---|---|
| ICM (Pathak et al., 2017) | **Embedding** + Forward dynamics + Inverse dynamics | $16, 074, 512 +2, 110, 464 + 527, 371$ | 73% |
| RND (Burda et al., 2019b) | **Embedding** | $16, 074, 512$ | 63% |
| RIDE (Burda et al., 2019b) | **Embedding** + Forward dynamics + Inverse dynamics | $16, 074, 512 +2, 110, 464 + 527, 371$ | 73% |
| NovelD (Zhang et al., 2021a) | **Embedding** | $16, 074, 512$ | 63% |
| E3B (Henaff et al., 2022; 2023) | **Embedding** + Inverse dynamics | $16, 074, 512 +527, 371$ | 65% |
| CAE | - | - | 0 |
| CAE+ | Inverse dynamics | $199, 819$ | 0.8% |

## 3  Methodology

Unless otherwise specified, bold uppercase symbols denote matrices, while bold lowercase symbols represent vectors. $\boldsymbol{I}$ refers to an identity matrix. Frobenius norm for matrices and $l_2$ norm for vectors are both denoted by $\|\cdot\|_2$. Mahalanobis norm of a vector $\boldsymbol{x}$ with respect to $\boldsymbol{A}$ is defined as $\|\boldsymbol{x}\|_{\boldsymbol{A}} = \sqrt{\boldsymbol{x}^\mathsf{T}\boldsymbol{A}\boldsymbol{x}}$. For any integer $K > 0$, the set of integers $\{1, 2, ..., K\}$ is denoted by $[K]$.

### 3.1  Preliminary

An episodic Markov Decision Process (MDP) is formally defined as a tuple $(\mathcal{S}, \mathcal{A}, H, \mathbb{P}, r)$, where $\mathcal{S}$ denotes the state space and $\mathcal{A}$ is the action space. Integer $H > 0$ indicates the duration of each episode. Functions $\mathbb{P} : \mathcal{S} \times \mathcal{A} \times \mathcal{S} \to [0, 1]$ and $r : \mathcal{S} \times \mathcal{A} \to [0, 1]$ are the Markov transition and reward functions, respectively. During an episode, the agent follows a policy $\pi : \mathcal{S} \times \mathcal{A} \to [0, 1]$. At each time step $h \in [H]$ in the episode, the agent observes the current state $s_h \in \mathcal{S}$ and selects an action $a_h \sim \pi(\cdot|s_h)$ to execute, then the environment transits to the next state $s_{h+1} \sim \mathbb{P}(\cdot|s_h, a_h)$, yielding an immediate reward $r_h = r(s_h, a_h)$. At time step $h$, the action-value function $Q(s_h, a_h)$ measures the cumulative return obtained by taking action $a_h$ in state $s_h$ and subsequently following policy $\pi$:

$$Q(s_h, a_h) = \sum_{t=h}^{H} \gamma^{t-h} r_t \tag{1}$$

where $0 \leq \gamma \leq 1$ is the discount parameter.

Many algorithms have been developed to learn the optimal policy $\pi^*$ for the agent to select and execute actions at each time step $h$ in the episode, thus ultimately maximizing the cumulative return $\sum_{h=1}^{H} \gamma^{h-1} r_h$. Notable algorithms include DQN (Mnih et al., 2015), PPO (Schulman et al., 2017), SAC (Haarnoja et al., 2018), TD3 (Fujimoto et al., 2018), IMPALA (Espeholt et al., 2018), DSAC (Duan et al., 2021; 2023), and others. A common component of these algorithms is a neural network that parameterizes the action-value

function[1] $Q(\cdot, \cdot)$ under a specific policy as Equation 2, where $\phi(\cdot, \cdot | \boldsymbol{W})$ is the embedding layers, $\boldsymbol{\theta}$ and $\boldsymbol{W}$ are trainable parameters of the network.

$$Q(s, a) = \boldsymbol{\theta}^{\mathsf{T}} \phi(s, a | \boldsymbol{W}) \tag{2}$$

Bellman loss as Equation 3 is often employed to learn the action-value function. Using the most recent action-value function, the policy can be updated in various ways, depending on the specific algorithm. Since CAE focuses on leveraging Equation 2 for efficient exploration while preserving the core techniques of existing RL algorithms, we introduce CAE within the context of DQN for simplicity. However, it can be easily adapted to other RL algorithms.

$$L_B = \left( Q(s_h, a_h) - \mathbb{E}_{s_{h+1} \sim \mathbb{P}(\cdot | s_h, a_h)} \left[ r_h + \gamma \cdot \max_{a_{h+1}} Q(s_{h+1}, a_{h+1}) \right] \right)^2 \tag{3}$$

## 3.2 CAE: the Critic as an Explorer

For a state-action pair $(s, a)$, the estimated action-value $Q(s, a)$ is subject to an uncertainty term $\beta(s, a)$, arising from the novelty or limited experience with the particular state-action pair. Similar to MAB problems, explicitly accounting for such uncertainty is crucial. Incorporating this uncertainty term promotes exploration and can lead to improved long-term returns. Accordingly, we adjust the action-value function with an uncertainty bonus, as shown in Equation 4, where $\alpha \geq 0$ denotes the exploration coefficient. In the literature, *uncertainty* is often also referred to as a *bonus*, and we use the two terms interchangeably when no ambiguity arises.

$$Q(s, a) = \boldsymbol{\theta}^{\mathsf{T}} \phi(s, a | \boldsymbol{W}) + \alpha \beta(s, a) \tag{4}$$

However, defining $\beta(s, a)$ remains challenging. Provably efficient methods often attempt to address this by either assuming a linear value function (Jin et al., 2018) or requiring $O(n^3)$ computation time (Yang et al., 2020) in terms of the number of parameters $n$ in the value network. Both of these approaches have drawbacks, *i.e.*, linearity fails to capture the complexity of tasks while the $O(n^3)$ computational cost is impractical.

To overcome these limitations, we draw inspiration from Neural-LinUCB (Xu et al., 2022) and Neural-LinTS (Riquelme et al., 2018), which effectively decouple representation learning from exploration strategies. Building on this idea and following the standard value network structure in Equation 2, CAE decomposes the action-value function into two distinct components.

- Network $\phi(s, a | \boldsymbol{W})$ extracts the embedding of the state-action pair $(s, a)$;

- $Q(s, a) = \boldsymbol{\theta}^{\mathsf{T}} \phi(s, a | \boldsymbol{W})$ is a linear function of the embedding $\phi(s, a | \boldsymbol{W})$ with parameter $\boldsymbol{\theta}$.

Consequently, after appropriate modifications, MAB theory under the linearity assumption can be adapted to work with the embeddings $\phi(s, a)$ for $\forall s \in \mathcal{S}$ and $\forall a \in \mathcal{A}$. Simultaneously, the action-value function retains its representational capacity through the embedding layers $\phi(s, a)$, ensuring promising practical performance. While various MAB techniques can be adapted to the embeddings, we illustrate CAE using the two most representative ones. Other techniques can be utilized similarly, showcasing that CAE is a generalizable framework rather than a fixed method.

**Upper Confidence Bound (UCB)** is an optimistic exploration strategy in MAB. It defines the uncertainty term as Equation 5, where $\phi(\cdot, \cdot)$ denotes the latest embedding layers, and $\boldsymbol{A}$ is the Gram matrix, initialized as $\boldsymbol{A} = \lambda \boldsymbol{I}$ with $\lambda$ being the ridge regularization parameter. At each step, $\boldsymbol{A}$ is updated using Equation 6.

---

[1]In some algorithms, the state-value function is learned instead of the action-value function. However, this does not affect the implementation and conclusion of our method, as will be seen in Subsection 3.2.

$$\beta(s,a) = \sqrt{\phi(s,a)^\mathsf{T} \boldsymbol{A}^{-1} \phi(s,a)} \tag{5}$$

$$\boldsymbol{A} \leftarrow \boldsymbol{A} + \boldsymbol{\phi}(s,a)\boldsymbol{\phi}^\mathsf{T}(s,a) \tag{6}$$

**Thompson Sampling** is a randomized exploration strategy that samples the value function, adjusted for uncertainty, from a posterior distribution. It defines the uncertainty term as Equation 7, where the Gram matrix $\boldsymbol{A}$ is initialized and updated in the same manner as in UCB.

$$\boldsymbol{\Delta\theta} \sim N(0, \boldsymbol{A}^{-1}); \quad \beta(s,a) = (\boldsymbol{\Delta\theta})^\mathsf{T} \phi(s,a) \tag{7}$$

As the value network undergoes continuous updates, exploration based on the ever-changing embedding layers $\phi(\cdot, \cdot)$ can become highly unstable, significantly impairing practical performance. Inspired by existing scaling strategies (Welford, 1962; Elsayed et al., 2024), we adopt an appropriate one for the generated uncertainty at each time step, ensuring both stability and practical functionality, as detailed in Algorithm 1. This scaling strategy normalizes the generated uncertainty at each time step using the running standard deviation, which, despite its simplicity, has a profound impact on the performance of CAE. The critical importance of this design is further highlighted through ablation studies presented in Appendix E.

**Adapting CAE to General RL Algorithms.** Depending on the RL algorithm employed, we may sometimes learn a state-value network instead of an action-value network. In such cases, the network produces embeddings for states rather than for state-action pairs. Even when an action-value network is learned, it might still output only state embeddings if it is designed to take a state as input and produce values for multiple actions. In these scenarios, we use either the embedding of the next state or the embedding concatenated with the action as a proxy for the current state–action embedding when computing uncertainty.

---

**Algorithm 1** Scaling strategy for the uncertainty

---

1: **Input:** Uncertainty $b$, running mean $\mu$, cumulative squared deviation $\nu^2$, and running count of samples $\mathcal{N}$
2: Update the sample count $\mathcal{N} \leftarrow \mathcal{N} + 1$
3: Compute $\delta = b - \mu$
4: Update the running mean $\mu \leftarrow \mu + \frac{\delta}{\mathcal{N}}$
5: Update the cumulative squared deviation $\nu^2 \leftarrow \nu^2 + \delta \times (b - \mu)$
6: **Output:** Scaled uncertainty $\frac{b}{\sqrt{\nu^2/\mathcal{N}}}$, and the updated $\mu$, $\nu^2$, and $\mathcal{N}$

---

### 3.3 CAE+: Enhancing CAE with Minimal Overhead

For tasks with complex dynamics or very sparse rewards, learning value networks is particularly challenging, which in turn hinders exploration reliant on them. Accordingly, we propose CAE+, an extension of CAE that incorporates a lightweight auxiliary network, introducing less than 1% additional parameters.

Specifically, we utilize an Inverse Dynamics Network (IDN) (Pathak et al., 2017; Raileanu & Rocktäschel, 2020; Henaff et al., 2022) to enhance the learning of the embedding layers contained in the value networks. This is achieved by a compact network $f$ that infers the distribution $p(a_h)$ over taken actions given consecutive states $s_h$ and $s_{h+1}$, which is trained by maximum likelihood estimation as Equation 8.

$$L_f = -\log p(a_h | s_h, s_{h+1}) \tag{8}$$

We utilize the embedding layers as follows:

- If the value network is an action-value network with embedding layers $\phi(s,a)$, a fixed default action $\ddot{a}$ is assigned to the action input while the true states are used; the resulting outputs of $\phi(\cdot, \ddot{a})$ are then treated as state embeddings.

- If the value network is a state-value network with embedding layers $\phi(s)$, the actual states are directly fed in to generate their embeddings.

These state embeddings are subsequently transformed by a linear layer $\boldsymbol{U}$, followed by a small network $\bar{f}$, which processes the transformed consecutive embeddings to infer the action. Equation 9 illustrates this procedure for the case of embedding layers $\phi(s, a)$.

$$p(a_h|s_h, s_{h+1}) = f(\phi(s_h, \ddot{a}), \phi(s_{h+1}, \ddot{a})) = \bar{f}(\boldsymbol{U}\phi(s_h, \ddot{a}), \boldsymbol{U}\phi(s_{h+1}, \ddot{a})) \tag{9}$$

Although incorporating the IDN loss can accelerate learning of the embedding layers by leveraging knowledge of environment dynamics, it also introduces additional constraints on the shared embedding space, thereby limiting the flexibility of the embeddings for exploration. In CAE+, to address this limitation, we adopt the transformed embeddings $\boldsymbol{U}\phi(s, a)$ instead of the original ones $\phi(s, a)$ to calculate the uncertainty. Consequently, by replacing $\phi(s, a)$ with $\boldsymbol{U}\phi(s, a)$, Equations 5 and 7, which are used in CAE to generate uncertainty, are modified into Equation 10, with the Gram matrix $\boldsymbol{A}$ updated according to Equation 11. Complete procedure of CAE+ is in Algorithm 2.

$$\beta(s, a) \triangleq \begin{cases} \sqrt{\phi^{\mathsf{T}}(s, a)\boldsymbol{U}^{\mathsf{T}}\boldsymbol{A}^{-1}\boldsymbol{U}\phi(s, a)} & \text{UCB} \\ (\boldsymbol{\Delta\theta})^{\mathsf{T}}\boldsymbol{U}\phi(s, a) \big|_{\boldsymbol{\Delta\theta} \sim \mathcal{N}(0, \boldsymbol{A}^{-1})} & \text{Thompson Sampling} \end{cases} \tag{10}$$

$$\boldsymbol{A} \leftarrow \boldsymbol{A} + \boldsymbol{U}\phi(s, a)\phi^{\mathsf{T}}(s, a)\boldsymbol{U}^{\mathsf{T}} \tag{11}$$

In CAE+, the structure of the network $f$ offers several advantages. First, the transformation $\boldsymbol{U}$ decouples environment dynamics from returns, mitigating interdependencies that could hinder flexibility and thereby enhancing empirical performance, as evidenced by the ablation studies presented later. Second, since $\boldsymbol{U}$ is a simple linear transformation, it approximately preserves the theoretical guarantees of UCB-based and Thompson Sampling-based exploration strategies, ensuring both rigor and stability in practice. Third, by projecting $\phi(s, a)$ into a lower-dimensional embedding with $\bar{d} < d$, the approach not only reduces the number of additional parameters but also lowers the computational complexity of uncertainty estimation at each time step, i.e., from $O(d^3)$ to $O(\bar{d}^3)$, making the method more efficient.

**Speed Up CAE+ with Rank$-1$ Update** According to Algorithm 2, the Gram matrix $\boldsymbol{A}$ needs to be inverted at each step, which is cubic in dimension. **Alternatively**, we can use the Sherman-Morrison matrix identity (Sherman & Morrison, 1950; Henaff et al., 2022) to perform rank$-1$ updates of $\boldsymbol{A}^{-1}$ in quadratic time as Equation 12.

$$\boldsymbol{A}^{-1} \leftarrow \boldsymbol{A}^{-1} - \frac{\boldsymbol{A}^{-1}\boldsymbol{U}\phi(s, a)\phi^{\mathsf{T}}(s, a)\boldsymbol{U}^{\mathsf{T}}(\boldsymbol{A}^{-1})^{\mathsf{T}}}{1 + \phi^{\mathsf{T}}(s, a)\boldsymbol{U}^{\mathsf{T}}\boldsymbol{A}^{-1}\boldsymbol{U}\phi(s, a)} \tag{12}$$

## 4 Theoretical Analysis

Under the optimal policy $\pi^*$, let the corresponding action-value function $Q^*$ be structured as in Equation 2 and parameterized by $\boldsymbol{\theta}^*$ and $\boldsymbol{W}^*$. In Algorithm 2, the policy executed in episode $m \in [M]$ is denoted by $\pi_m$, and its associated action-value function is $Q^{\pi_m}$. Cumulative regret of Algorithm 2 is given in Definition 1.

**Definition 1.** *Cumulative Regret. After $M$ episodes of interactions with the environment, the cumulative regret of CAE or CAE+ is defined as Equation 13, where $u_1^m$ is the optimal action at state $s_1^m$ generated by policy $\pi^*$ while $a_1^m$ is that selected by the executed policy $\pi_m$.*

$$\mathrm{R}_M = \sum_{m=1}^{M} Q^*(s_1^m, u_1^m) - Q^{\pi_m}(s_1^m, a_1^m) \tag{13}$$

---

**Algorithm 2** CAE+ with Action-value Network

---
1: **Input:** Ridge parameter $\lambda > 0$, exploration parameter $\alpha \geq 0$, episode length $H$, episode number $M$, learning rate $\eta$
2: **Initialize:** Gram matrix $\boldsymbol{A} = \lambda \boldsymbol{I}$, initial policy $\pi(\cdot)$ and value function $Q(\cdot, \cdot)$, network $f(\cdot|\boldsymbol{U})$, running mean $\mu = 0$, cumulative squared deviation $\nu^2 = 0$, running count $\mathcal{N} = 0$
3: **for** episode $m = 1$ **to** $M$ **do**
4:     Receive the initial state $s_1^m$ from the environment
5:     **for** step $h = 1, 2, ..., H - 1$ **do**
6:         Execute action $a_h^m \sim \pi(s_h^m)$, observe the next state $s_{h+1}^m$, and receive the immediate reward $r_h^m$
7:         Generate bonus $\beta(s_h^m, a_h^m)$ by Equation 10
8:         Provide $\beta(s_h^m, a_h^m)$, $\mu$, $\nu^2$, and $\mathcal{N}$ as inputs to Algorithm 1 to obtain scaled bonus $\beta_h^m$ and updated $\mu$, $\nu^2$, and $\mathcal{N}$
9:         Reshape the reward $\tilde{r}_h^m = r_h^m + \alpha \beta_h^m$
10:        Update the Gram matrix $\boldsymbol{A}$ by Equation 11
11:     **end for**
12:     Sample a batch $\mathcal{B} = \left\{ s_h^t, a_h^t, s_{h+1}^t, \tilde{r}_h^t \mid h \in [1, H-1], t \leq m \right\}$
13:     Calculate the IDN loss $L_f$ by Equation 8 and the Bellman loss $L_B$ by Equation 3 on batch $\mathcal{B}$
14:     Update value function $Q(\cdot, \cdot)$ and network $f$ jointly by minimizing the combined loss $\min(L_f + L_B)$ with step size $\eta$
15:     Update the policy $\pi(\cdot)$ based on the latest value function $Q(\cdot, \cdot)$
16: **end for**

---

The cumulative regret measures the gap between the optimal return and the actual return accumulated over $M$ episodes. As discussed earlier, CAE draws inspiration from Neural-LinUCB (Xu et al., 2022) and Neural-LinTS (Riquelme et al., 2018). While Neural-LinUCB is supported by theoretical guarantees, Neural-LinTS has so far only been validated empirically. In this work, we complete the regret analysis for Neural-LinTS and subsequently derive the regret bound of CAE. Before stating Theorem 1, we introduce the standard assumptions used in the literature on *deep representation and shallow exploration* (Xu et al., 2022) as Assumption 1 and Assumption 2. Assumption 3 is deferred to the Appendix because it requires more detailed notation.

**Assumption 1.** *Assume that $\|(s; a)\|_2 = 1$ for $\forall s \in \mathcal{S}, \forall a \in \mathcal{A}$. The entries of $(s; a)$ satisfy Equation 14, where $j = 1, 2, \ldots, \frac{D}{2}$ and $D$ represents the dimension of $(s; a)$.*

$$(s; a)_j = (s; a)_{j + \frac{D}{2}} \tag{14}$$

Note that even if the original state-action pairs $(s; a)$ do not satisfy this assumption, they can be preprocessed by augmenting them to $(s; a; s; a)$ and applying appropriate scaling to ensure the assumption holds.

**Assumption 2.** *The neural tangent kernel $\boldsymbol{H}$ of the action-value network is positive definite.*

Neural tangent kernel $\boldsymbol{H}$ is defined in accordance with a recent line of research (Jacot et al., 2018; Arora et al., 2019; Xu et al., 2022) and is essential for the analysis of overparameterized neural networks. A detailed discussion of these assumptions is deferred to Appendix D.3, where we show that they are mild and commonly adopted in the literature. In the following, we present the regret guarantee of CAE.

**Theorem 1.** *Assume that Assumption 1, Assumption 2, and Assumption 3 hold, and that $\|\boldsymbol{\theta}^*\|_2 \leq 1$ as well as $\|\phi(s, a)\|_2 \leq 1$ for $\forall s \in \mathcal{S}, \forall a \in \mathcal{A}$. For any $\sigma \in (0, 1)$, assume that the number of parameters $\iota$ in each of the $L$ layers of $\phi(\cdot, \cdot)$ satisfies $\iota = \text{poly}(L, D, \frac{1}{\sigma}, \log \frac{M|\mathcal{A}|}{\sigma})$, where $|\mathcal{A}|$ denotes the size of the action space and $\text{poly}(\cdot)$ denotes a polynomial function of the listed variables. Set the exploration coefficient and step size as:*

$$\alpha = \sqrt{2(D \cdot \log(1 + \frac{M \cdot \log |\mathcal{A}|}{\lambda}) - \log \sigma)} + \sqrt{\lambda}$$

$$\eta \leq C_1 (\iota \cdot D^2 M^{\frac{11}{2}} L^6 \cdot \log \frac{M|\mathcal{A}|}{\sigma})^{-1}$$

*then with probability at least $1 - \sigma$, it holds that:*

$$\mathrm{R}_M \leq C_2\alpha H\sqrt{MD\log(1+\frac{M}{\lambda D})} + C_4 H\sqrt{MH\log\frac{1}{\sigma}} + \frac{C_3 H L^3 D^{\frac{5}{2}} M \sqrt{\log\iota \log\left(\frac{1}{\sigma}\right)\log\left(\frac{M|\mathcal{A}|}{\sigma}\right)}\,\|\boldsymbol{q}-\tilde{\boldsymbol{q}}\|_{\boldsymbol{H}^{-1}}}{\iota^{\frac{1}{6}}}$$

*where $C_1, C_2, C_3, C_4$ are constants; $\boldsymbol{q}$ and $\tilde{\boldsymbol{q}}$ are the target value vector and the estimated value vector of the action-value network, respectively. More discussions of these notations are in Appendix B and Appendix D.*

Specifically, we assume $\|\boldsymbol{\theta}^*\|_2 \leq 1$ and $\|\phi(s,a)\|_2 \leq 1$ to make the bound scale-free. From this theorem, we can conclude that the upper bound of the cumulative regret grows sub-linearly with the number of episodes $M$, *i.e.*, $\widetilde{O}(\sqrt{M})$ where $\widetilde{O}(\cdot)$ hide constant and logarithmic dependence of $M$, indicating that the executed policy improves over time. Since MAB techniques are applied to the linear layer on top of the embedding layers, the cumulative regret naturally consists of two components, i.e., the exploration regret from the linear layer and the error induced by the network's estimation, which appears as the last term in the regret bound. It involves a trade-off between $M$ and $\iota$. Moreover, as the estimation error $\|\boldsymbol{q}-\tilde{\boldsymbol{q}}\|_{\boldsymbol{H}^{-1}}$ decreases over time, this term typically becomes negligible.

## 5 Experiment

**Benchmarks.** In our experiments, we evaluate CAE and CAE+ on MuJoCo, MiniHack, and Habitat, which correspond to dense-reward, sparse-reward, and reward-free settings, respectively.

**Baselines.** We evaluate our approach against **nine** baselines, *i.e.*, SAC (Haarnoja et al., 2018), PPO (Schulman et al., 2017), TD3 (Fujimoto et al., 2018), DSAC (Duan et al., 2021; 2023), ICM (Pathak et al., 2017), RND (Burda et al., 2019b), RIDE (Raileanu & Rocktäschel, 2020), NovelD (Zhang et al., 2021a), and E3B (Henaff et al., 2022; 2023).

For **MuJoCo** tasks, we evaluate SAC, PPO, TD3, and DSAC, both with and without CAE. Notably, the other baselines are excluded from the MuJoCo experiments, as they are rarely applied to dense reward settings. Since CAE introduces no additional parameters, it is meaningful to assess whether it can improve existing RL algorithms without increasing training overhead.

For **MiniHack and Habitat** tasks, we adopt IMPALA (Espeholt et al., 2018) and PPO as the base RL algorithms, respectively, following the standard configurations in the open-source codebases. We compare CAE and CAE+ against ICM, RND, RIDE, NovelD, and E3B. Since E3B achieves state-of-the-art performance on both MiniHack and Habitat, we report only the results of E3B. For results of ICM, RND, RIDE, and NovelD, refer to the E3B paper (Henaff et al., 2022; 2023).

All the experiments are based on open-source codebases from E3B, CleanRL (Huang et al., 2022), DSAC, and Habitat-lab (Savva et al., 2019; Andrew et al., 2021; Xavi et al., 2023). The core code and hyperparameters are provided in Appendices A and E, respectively. All experiments were conducted on an Ubuntu 22.04 LTS system equipped with a 13th Gen Intel Core i9-13900KF CPU and an NVIDIA RTX 4090 GPU.

### 5.1 MuJoCo tasks with Dense Rewards

Table 3: Experimental results after $1e6$ environment interaction steps on MuJoCo-v4 tasks, except for the *Humanoid* task, which is evaluated after $4e6$ steps. *RPI* represents the **R**elative **P**erformance **I**mprovement achieved by CAE. Refer to Appendix E for the corresponding experimental curves.

| Alg. \ Env | PPO | PPO + CAE | *RPI %* | TD3 | TD3 + CAE | *RPI %* | SAC | SAC + CAE | *RPI %* |
|---|---|---|---|---|---|---|---|---|---|
| Swimmer | $99 \pm 11.5$ | $\mathbf{107 \pm 6.47}$ | 8.08 | $78 \pm 15.4$ | $\mathbf{130 \pm 14.2}$ | 66.7 | $61 \pm 35.2$ | $\mathbf{161 \pm 26.9}$ | 164 |
| Hopper | $\mathbf{2503 \pm 786.6}$ | $2453 \pm 673.2$ | $-2.00$ | $3044 \pm 574.0$ | $\mathbf{3244 \pm 226.3}$ | 6.57 | $2908 \pm 600.8$ | $\mathbf{3188 \pm 485.2}$ | 9.63 |
| Walker2d | $3405 \pm 842.0$ | $\mathbf{3554 \pm 928.0}$ | 4.38 | $3764 \pm 234.4$ | $\mathbf{4251 \pm 567.1}$ | 12.9 | $4362 \pm 405.5$ | $\mathbf{4742 \pm 484.4}$ | 8.71 |
| Ant | $1762 \pm 540.0$ | $\mathbf{2378 \pm 843.4}$ | 35.0 | $3492 \pm 1745.7$ | $\mathbf{5074 \pm 519.3}$ | 45.3 | $4846 \pm 1306.4$ | $\mathbf{5482 \pm 511.9}$ | 13.1 |
| HalfCheetah | $2636 \pm 1344.3$ | $\mathbf{3104 \pm 926.0}$ | 17.8 | $10316 \pm 193.8$ | $\mathbf{10473 \pm 563.4}$ | 1.52 | $11154 \pm 457.1$ | $\mathbf{11587 \pm 418.3}$ | 3.88 |
| Humanoid | $619 \pm 92.1$ | $\mathbf{646 \pm 127.1}$ | 4.36 | $5973 \pm 257.7$ | $\mathbf{6275 \pm 483.2}$ | 5.06 | $\mathbf{5261 \pm 186.4}$ | $5218 \pm 228.4$ | $-0.82$ |
| *RPI Mean* | - | - | 11.3 | - | - | 23.0 | - | - | 33.1 |

MuJoCo testbed is a widely used physics-based simulation environment. MuJoCo provides a suite of continuous control tasks where agents must learn to perform various actions, such as locomotion, manipulation, and balancing, within simulated robotic environments. Since comparisons among state-of-the-art RL baselines, such as PPO, SAC, TD3, and DSAC on MuJoCo, have been extensively covered in previous studies, our focus is on investigating how CAE can enhance these algorithms.

Results are summarized in Table 3, averaged over random seeds $\{1, 2, 3, 4, 5\}$. The results show that CAE consistently improves the performance of PPO, TD3, and SAC across most MuJoCo tasks. Notably, TD3 and SAC augmented with CAE achieve substantially better performance on the *Swimmer* task. Although this task is not considered particularly challenging, the standalone TD3 and SAC baselines obtain relatively limited returns.

In Table 4, we summarize the performance of DSAC with and without CAE. These experiments are conducted on MuJoCo-v3, rather than MuJoCo-v4, because the open-source DSAC codebase is built upon MuJoCo-v3. As shown, incorporating CAE consistently improves the performance of DSAC across the MuJoCo benchmark tasks.

Table 4: Experimental results after $1e6$ interaction steps on MuJoCo-v3, except for the *Humanoid* task, which is evaluated after $2e6$ steps.

| Alg. Env | DSAC | DSAC + CAE | *RPI %* |
|---|---|---|---|
| Swimmer | $131 \pm 14.8$ | $\mathbf{150 \pm 7.96}$ | 14.5 |
| Hopper | $2417 \pm 541.6$ | $\mathbf{2845 \pm 594.5}$ | 17.7 |
| Walker2d | $5550 \pm 624.0$ | $\mathbf{6069 \pm 422.1}$ | 9.35 |
| Ant | $5912 \pm 809.7$ | $\mathbf{6305 \pm 322.7}$ | 6.65 |
| HalfCheetah | $16036 \pm 439.1$ | $\mathbf{16338 \pm 249.1}$ | 1.88 |
| Humanoid | $10059 \pm 996.1$ | $\mathbf{10333 \pm 1104.4}$ | 2.72 |
| *RPI Mean* | - | - | 8.80 |

## 5.2 MiniHack tasks with Sparse Rewards

MiniHack (Samvelyan et al., 2021) is built on the NetHack Learning Environment (Küttler et al., 2020), a challenging video game where an agent navigates procedurally generated dungeons to retrieve a magical amulet. MiniHack tasks present a diverse set of challenges, such as locating and utilizing magical objects, traversing hazardous environments like lava and monsters. These tasks are characterized by sparse rewards, and the state provides a wealth of information, including images, texts, and more, though only a subset is relevant to each specific task.

As shown in Table 2, CAE+ introduces only a 0.8% increase in parameters compared to the base RL algorithm, IMPALA. In contrast, other exploration

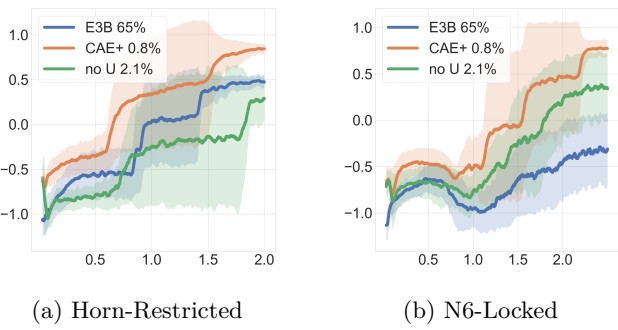

(a) Horn-Restricted     (b) N6-Locked

Figure 2: Ablation study to $U$ on MiniHack. Horizontal axis denotes the steps in multiples of $1e7$.

baselines, such as RIDE and E3B, require $60\% - 80\%$ additional parameters, underscoring the lightweight design of CAE+. Experimental results for E3B and CAE+, averaged over seeds $\{1, 2, 3\}$, are summarized in Table 5. Performance is evaluated on nine representative tasks, *i.e.*, five *navigation* tasks and four *skill* tasks. The results demonstrate that CAE+ consistently outperforms E3B across tasks. In particular, on challenging tasks such as *N6-Locked* and *LavaCross*, CAE+ achieves substantial performance improvements of 348% and 282%, respectively, highlighting its strong exploration capabilities.

**Ablation study to CAE on MiniHack.** Results of CAE on MiniHack tasks are provided in Table 5. As shown, CAE successfully solves a subset of tasks and even surpasses CAE+ in certain cases. However, it struggles to achieve positive performance in others, such as *N6-Locked*, *etc.*, which pose significant exploration challenges. This limitation stems from the difficulty of training effective value networks in complex environments, adversely affecting exploration reliant on them.

**Ablation study to the transformation $U$.** Additionally, we present experimental results for CAE+ without the transformation matrix $U$ in the auxiliary network $f$. As shown in Figure 2, CAE+ without $U$ occasionally outperforms E3B, though there are instances where it does not. Importantly, it consistently

underperforms the full CAE+ method with $U$. Moreover, CAE+ introduces more additional parameters without matrix $U$, specifically 2.1%.

**Wall-clock running time analysis.** RL experiments involve both network training and environment interaction on CPUs; therefore, wall-clock time is typically not regarded as a standard evaluation metric. Nevertheless, to provide a clear picture of the practical efficiency of CAE, we report wall-clock running times on MiniHack tasks. CAE requires approximately 17 hours to complete training per task, whereas E3B takes around 22 hours, demonstrating the superior efficiency of CAE.

Table 5: Experimental results after $2e7 - 3e7$ interaction steps on nine MiniHack tasks, whose detailed descriptions are in Appendix E.2. *RPI* quantifies the improvement of CAE+ compared to E3B. Since E3B is the state-of-the-art on MiniHack (Henaff et al., 2022), and CAE+ consistently outperforms it, we conclude that **CAE+ ≻ E3B, ICM, RND, RIDE, NovelD**. Refer to Appendix E for experimental figures.

| Env
Alg. | N4 | N4-Locked | N6 | N6-Locked | N10-OD | Horn | Random | Wand | LavaCross |
|---|---|---|---|---|---|---|---|---|---|
| E3B | $0.86 \pm 0.010$ | $0.72 \pm 0.090$ | $0.75 \pm 0.019$ | $-0.31 \pm 0.403$ | $0.71 \pm 0.042$ | $0.47 \pm 0.071$ | $0.57 \pm 0.071$ | $0.49 \pm 0.201$ | $0.22 \pm 0.412$ |
| CAE | $0.93 \pm 0.014$ | $0.84 \pm 0.041$ | $-0.46 \pm 0.193$ | $-0.37 \pm 0.310$ | $-0.84 \pm 0.216$ | $\mathbf{0.92 \pm 0.022}$ | $\mathbf{0.93 \pm 0.022}$ | $\mathbf{0.93 \pm 0.030}$ | $0.16 \pm 0.394$ |
| CAE+ | $\mathbf{0.97 \pm 0.006}$ | $\mathbf{0.87 \pm 0.017}$ | $\mathbf{0.94 \pm 0.014}$ | $\mathbf{0.77 \pm 0.093}$ | $\mathbf{0.86 \pm 0.023}$ | $0.84 \pm 0.055$ | $0.80 \pm 0.040$ | $0.65 \pm 0.131$ | $\mathbf{0.84 \pm 0.024}$ |
| *RPI %* | 12.79 | 20.83 | 25.3 | 348.39 | 21.13 | 78.72 | 40.35 | 32.65 | 281.82 |

### 5.3 Reward-free Habitat task

Habitat (Savva et al., 2019; Andrew et al., 2021; Xavi et al., 2023) is a platform for embodied AI research that supports agent navigation and interaction within simulations of real-world indoor environments. The experiments in this subsection are designed to evaluate exploration capabilities in visually rich, realistic settings. We employ the HM3D dataset (Santhosh et al., 2021), which comprises high-quality reconstructions of $1,000$ diverse indoor spaces. Agents are trained **solely** with the generated exploration bonuses on the *PointNav* task. Evaluation is performed in unseen test environments by measuring the proportion of the environment revealed.

Results for E3B, CAE, and CAE+, averaged over three seeds $\{1, 2, 3\}$, are summarized in Table 6, highlighting the superior performance of CAE+. Since E3B is the state-of-the-art on Habitat, and CAE+ further outperforms E3B, it follows that **CAE+ ≻ E3B, ICM, RND, RIDE, NovelD**. Figure 3 provides an illustration of a specific case,

Table 6: Experimental results after $1e8$ interaction steps on Habitat.

| E3B | CAE | CAE+ | *RPI %* |
|---|---|---|---|
| $0.51 \pm 0.097$ | $0.49 \pm 0.102$ | $\mathbf{0.69 \pm 0.074}$ | 35.29 |

where the trained policy of CAE+ explores a larger portion of the environment compared to that of E3B.

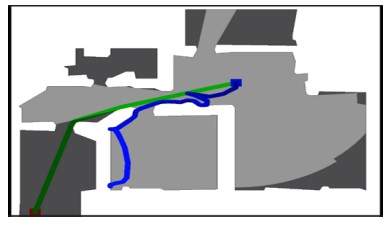
(a) CAE+ reveals 0.67 of the map

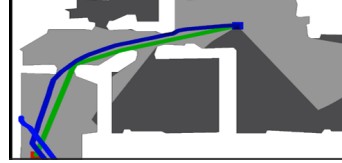
(b) E3B reveals 0.53 of the map

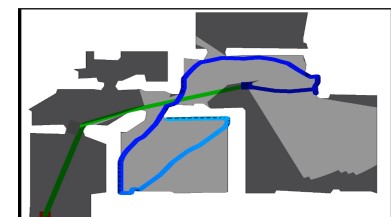
(c) CAE reveals 0.51 of the map

Figure 3: Trajectories of the learned policies on a Habitat environment unseen during training.

## 6 Conclusion

In this paper, we propose CAE, a lightweight exploration method that seamlessly integrates with existing RL algorithms without adding parameters. CAE exploits the value network's embedding layers to guide

exploration, requiring no changes to the rest of the algorithm. A simple scaling strategy ensures stable learning. For sparse-reward tasks, we extend CAE to CAE+ by adding a small auxiliary network, accelerating learning with minimal overhead. We provide theoretical guarantees in the form of sub-linear regret bounds. Extensive experiments show that CAE and CAE+ significantly outperform baseline methods across dense-reward, sparse-reward, and reward-free settings.

## Broader Impact Statement

CAE and CAE+ bridge the gap between provably efficient and practically successful exploration methods. While theoretically grounded approaches often suffer from scalability and applicability issues, empirically driven methods typically lack theoretical guarantees and require extensive parameter training. Inspired by the *deep representation and shallow exploration* paradigm, this work proposes a novel framework that decomposes value networks in deep RL and repurposes their embedding layers for exploration, thereby achieving theoretical guarantees with minimal or no additional parameter training.

This framework contributes to the development of sample-efficient and interpretable exploration methods in deep RL, potentially accelerating progress in applications such as robotics, recommendation systems, autonomous decision-making systems, and large language models. Moreover, the lightweight nature of the method makes it suitable for deployment in resource-constrained settings. Future research could focus on integrating a broader spectrum of MAB techniques, evaluating the robustness across a wide range of tasks, and improving the computational efficiency of uncertainty estimation.

## Acknowledgments

The first author thanks Ping Huang and Hanfang Zhang for their assistance with configuring several open-source baselines, and Shuo Chen and Siyuan Qi for their substantial help in revising and improving the manuscript.

This work is supported by the National Natural Science Foundation of China (No. 62506041).

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

## A  Implementation

Our experiments are based on the following open-source codebases:

- E3B (Henaff et al., 2022): https://github.com/facebookresearch/e3b

- CleanRL (Huang et al., 2022): https://github.com/vwxyzjn/cleanrl

- DSAC (Duan et al., 2021; 2023): https://github.com/Jingliang-Duan/DSAC-v2

- Habitat-lab (Andrew et al., 2021; Xavi et al., 2023): https://github.com/facebookresearch/habitat-lab

In Listing 1, we present the core code of CAE, while the rest of the deep RL algorithm remains unchanged. As shown, CAE is simple to implement, integrates seamlessly with any existing RL algorithm, and requires no additional parameter learning beyond what is already in the RL algorithm.

Listing 1: CAE core code

```
A_inverse = torch.inverse(A)
phi = q_net.get_emb(torch.Tensor(obs), torch.Tensor(action)).squeeze().detach()
bouns = np.sqrt(torch.matmul(phi.T, torch.matmul(A_inverse, phi)).item())
reward += scaled_bonus
A += torch.outer(phi, phi)
```

In Listing 2, we present the additional code of CAE+ alongside that of CAE. As shown, CAE+ minimizes an additional loss, specifically the Inverse Dynamics Network (IDN) loss, in addition to the losses from the original RL algorithm.

Listing 2: Additional core code of CAE+

```
phi = q_net.get_emb(torch.Tensor(batch['obs']), torch.Tensor(default_actions))
predict_action = inverse_dynamic_net(phi[:-1], phi[1:])
idn_loss = compute_inverse_dynamics_loss(predict_action, batch['action'][:-1])

def compute_inverse_dynamics_loss(action, true_action):
    loss=F.nll_loss(F.log_softmax(action, dim=-1), true_action, reduction='none')
    return loss
```

# B  Long version of CAE

For a more comprehensive theoretical analysis, we present the theoretical version of CAE in Algorithm 3, where, for conciseness, we denote the embedding of the state-action pair at time step $h$ in episode $m$ by $\phi_h^m = \phi(s_h^m, a_h^m | \boldsymbol{W}_h^m)$. Following the standard notation in the literature on provable algorithms (Jin et al., 2020; Yang et al., 2020), function parameters are assumed to be independent across different time steps $h \in [H]$, and this convention is also adopted in Algorithm 3. As shown in the algorithm, the parameters $\boldsymbol{\theta}_h$ and $\boldsymbol{W}_h$ are updated iteratively to learn the two decomposed components of the action-value function in Equation 2 via the Bellman equation. Specifically, $\boldsymbol{\theta}_h$ is updated in Line 9 using its closed-form solution (Li et al., 2010), while the embedding network $\phi(\cdot, \cdot \mid \boldsymbol{W}_h)$ is kept fixed. Subsequently, in Line 10, the embedding network $\phi(s, a \mid \boldsymbol{W}_h)$ is updated with $\boldsymbol{\theta}_h$ held fixed. In this step, $\eta$ denotes the learning rate, $L_h^m$ is the Bellman loss, and $s_h^t, a_h^t, r_h^t$ for $\forall t \in [m]$ denote the collected historical experiences.

---

**Algorithm 3** DQN (Mnih et al., 2015) enhanced with CAE

---

1: **Input:** Ridge parameter $\lambda > 0$, exploration parameter $\alpha \geq 0$, episode length $H$, episode number $M$, step size $\eta$, and the discount factor $\gamma$

2: **Initialize:** Gram matrix $\boldsymbol{A}_h^1 = \lambda \boldsymbol{I}$, $\boldsymbol{b}_h^1 = \boldsymbol{0}$, parameters $\boldsymbol{\theta}_h^1 \sim \frac{1}{D} N(\boldsymbol{0}, \boldsymbol{I})$, networks $\phi(\cdot, \cdot | \boldsymbol{W}_h^1)$ (Xu et al., 2022), $Q_h^1 = (\boldsymbol{\theta}_h^1)^\mathsf{T} \phi(\cdot, \cdot | \boldsymbol{W}_h^1)$, and the target value-networks $\bar{Q}_h^1 = Q_h^1$, for $\forall h \in [H]$

3: **for** episode $m = 1$ **to** $M$ **do**

4:     Sample the initial state of the episode $s_1^m$

5:     **for** step $h = 1, 2, ..., H$ **do**

6:         Execute action $a_h^m = \arg\max_a Q_h^m(s_h^m, a)$ and get the next state $s_{h+1}^m$ and reward $r_h^m$

7:         Compute the target value $q_h^m = r_h^m + \gamma \cdot \max_a \bar{Q}_{h+1}^m(s_{h+1}^m, a)$

8:         Update $\boldsymbol{A}_h^{m+1} = \boldsymbol{A}_h^m + \phi_h^m (\phi_h^m)^\mathsf{T}$ and $\boldsymbol{b}_h^{m+1} = \boldsymbol{b}_h^m + q_h^m \phi_h^m$

9:         Update parameter $\boldsymbol{\theta}_h^{m+1} = (\boldsymbol{A}_h^{m+1})^{-1} \boldsymbol{b}_h^{m+1}$

10:         Update the embedding network to $\phi(\cdot, \cdot | \boldsymbol{W}_h^{m+1})$ by performing a gradient step $\boldsymbol{W}_h^{m+1} = \boldsymbol{W}_h^m - \eta \nabla_{\boldsymbol{W}_h^m} L_h^m$ where

$$L_h^m = \sum_{t=1}^m \left| (\boldsymbol{\theta}_h^{m+1})^\mathsf{T} \phi(s_h^t, a_h^t | \boldsymbol{W}_h^m) - r_h^t - \gamma \cdot \max_a \bar{Q}_{h+1}^m(s_{h+1}^t, a) \right|^2$$

11:         Obtain UCB-based uncertainty

$$\color{red} \beta_h^{m+1}(\cdot, \cdot) = \sqrt{(\phi(\cdot, \cdot | \boldsymbol{W}_h^{m+1}))^\mathsf{T} (\boldsymbol{A}_h^{m+1})^{-1} \phi(\cdot, \cdot | \boldsymbol{W}_h^{m+1})}$$

12:         Obtain Thompson Sampling-based uncertainty

$$\color{blue} \Delta\boldsymbol{\theta}_h^{m+1} \sim N(0, (\boldsymbol{A}_h^{m+1})^{-1}) \Longrightarrow \beta_h^{m+1}(\cdot, \cdot) = (\Delta\boldsymbol{\theta}_h^{m+1})^\mathsf{T} \phi(\cdot, \cdot | \boldsymbol{W}_h^{m+1})$$

13:         Approximate the action-value function

$$Q_h^{m+1}(\cdot, \cdot) = (\boldsymbol{\theta}_h^{m+1})^\mathsf{T} \phi(\cdot, \cdot | \boldsymbol{W}_h^{m+1}) + \alpha \beta_h^{m+1}(\cdot, \cdot)$$

14:     **end for**

15:     Update the target network $\bar{Q}_h^{m+1}(\cdot, \cdot) = Q_h^{m+1}(\cdot, \cdot), h \in [H]$

16: **end for**

---

Notably, $Q_h^m(s_h^m, a_h^m)$ denotes the estimated value, whereas $q_h^m$ represents the corresponding target value at each time step $h$ in episode $m$. By concatenating the values over all time steps $h \in [H]$ and episodes $m \in [M]$, we construct the estimated value vector $\tilde{\boldsymbol{q}}$ and the target value vector $\boldsymbol{q}$ of the action-value network, as referenced in Theorem 1.

## C Proof of Theorem 1

In this section, we analyze the cumulative regret bound of Algorithm 3. As $U$ is a straightforward linear transformation of the embeddings, it approximately preserves the theoretical guarantees of UCB-based and Thompson Sampling-based exploration strategies, thereby maintaining the rigor of CAE+.

Before delving into the detailed theory, we first review the notation used in this appendix. Let $\pi^*$ denote the true optimal policy, and let $\pi_m$ denote the policy executed in episode $m \in [M]$. More generally, for any policy—illustrated here with $\pi_m$—the relationship between the action-value function $Q^{\pi_m}$ and the corresponding maximum return $V^{\pi_m}$ can be expressed by:

$$V_h^{\pi_m}(s) = \max_a Q_h^{\pi_m}(s, a) \tag{15}$$

$$Q_h^{\pi_m}(s, a) = r_h(s, a) + \mathbb{E}_{s_{h+1} \sim \mathbb{P}_h(\cdot|s,a)} V_{h+1}^{\pi_m}(s_{h+1}) \tag{16}$$

In Algorithm 3, the estimated action-value function at step $h$ in episode $m$ is denoted by $Q_h^m(s, a)$, with the corresponding state-value function represented as $V_h^m(s)$. For clarity of presentation, we introduce the following additional notations.

$$(\mathbb{P}_h V_{h+1}^m)(s_h^m, a_h^m) = \mathbb{E}_{s_{h+1}^m \sim \mathbb{P}_h(\cdot|s_h^m, a_h^m)} V_{h+1}^m(s_{h+1}^m) \tag{17}$$

$$\delta_h^m(s_h^m, a_h^m) = r_h^m + (\mathbb{P}_h V_{h+1}^m)(s_h^m, a_h^m) - Q_h^m(s_h^m, a_h^m) \tag{18}$$

$$\zeta_h^m = \left[ V_h^m(s_h^m) - V_h^{\pi_m}(s_h^m) \right] - \left[ Q_h^m(s_h^m, a_h^m) - Q_h^{\pi_m}(s_h^m, a_h^m) \right] \tag{19}$$

$$\varepsilon_h^m = \left[ (\mathbb{P}_h V_{h+1}^m)(s_h^m, a_h^m) - (\mathbb{P}_h V_{h+1}^{\pi_m})(s_h^m, a_h^m) \right] - \left[ V_{h+1}^m(s_{h+1}^m) - V_{h+1}^{\pi_m}(s_{h+1}^m) \right] \tag{20}$$

Specifically, $\delta_h^m(s_h^m, a_h^m)$ represents the temporal-difference error for the state-action pair $(s_h^m, a_h^m)$. The notations $\zeta_h^m$ and $\varepsilon_h^m$ capture two sources of randomness, *i.e.*, the selection of action $a_h^m \sim \pi_m(\cdot|s_h^m)$ and the generation of the next state $s_{h+1}^m \sim \mathbb{P}_h(\cdot|s_h^m, a_h^m)$ from the environment.

*Proof.* **Theorem 1**.

Based on Lemma 1, the cumulative regret in Equation 13 can be decomposed into three terms as follows, where $\langle \cdot, \cdot \rangle$ means the inner product of two vectors.

$$\begin{aligned}
\mathrm{R}_M &= \sum_{m=1}^{M} Q_1^*(s_1^m, u_1^m) - Q_1^{\pi_m}(s_1^m, a_1^m) \\
&= \sum_{m=1}^{M} \sum_{h=1}^{H} \left[ \mathbb{E}_{\pi^*} \left[ \delta_h^m(s_h, a_h) | s_1 = s_1^m \right] - \delta_h^m(s_h^m, a_h^m) \right] + \sum_{m=1}^{M} \sum_{h=1}^{H} (\zeta_h^m + \varepsilon_h^m) \\
&\quad + \sum_{m=1}^{M} \sum_{h=1}^{H} \mathbb{E}_{\pi^*} \left[ \langle Q_h^m(s_h, \cdot), \pi_h^*(\cdot|s_h) - \pi_m(\cdot|s_h) \rangle | s_1 = s_1^m \right]
\end{aligned}$$

According to the definition of $\pi_m$, there is Equation 21.

$$\langle Q_h^m(s_h, \cdot), \pi_h^*(\cdot|s_h) - \pi_m(\cdot|s_h) \rangle \leq 0 \tag{21}$$

Consequently, with probability at least $1 - \sigma$ for $\sigma \in (0, 1)$, the cumulative regret in Equation 13 can be bounded as follows.

$$
\begin{aligned}
\text{R}_M &\leq \sum_{m=1}^{M}\sum_{h=1}^{H}\left[\mathbb{E}_{\pi^*}\left[\delta_h^m(s_h, a_h)|s_1 = s_1^m\right] - \delta_h^m(s_h^m, a_h^m)\right] + \sum_{m=1}^{M}\sum_{h=1}^{H}(\zeta_h^m + \varepsilon_h^m) \\
&\leq \sum_{m=1}^{M}\sum_{h=1}^{H}\left[\mathbb{E}_{\pi^*}\left[\delta_h^m(s_h, a_h)|s_1 = s_1^m\right] - \delta_h^m(s_h^m, a_h^m)\right] + \sqrt{16MH^3\log\frac{2}{\sigma_1}} \\
&\leq 4H\sqrt{MH\log\frac{2}{\sigma_2}} + C_2\alpha H\sqrt{MD\cdot\log(1+\frac{M}{\lambda D})} \\
&\quad + \frac{C_3\cdot HL^3D^{\frac{5}{2}}M\sqrt{\log\iota\log\left(\frac{1}{\sigma_2}\right)\log\left(\frac{M|\mathcal{A}|}{\sigma_2}\right)}\|\boldsymbol{q} - \tilde{\boldsymbol{q}}\|_{\boldsymbol{H}^{-1}}}{\iota^{\frac{1}{6}}} + \sqrt{16MH^3\log\frac{2}{\sigma_1}} \\
&\leq C_4 H\sqrt{MH\log\frac{1}{\sigma}} + C_2\alpha H\sqrt{MD\cdot\log(1+\frac{M}{\lambda D})} + \frac{C_3HL^3D^{\frac{5}{2}}M\sqrt{\log\iota\log\left(\frac{1}{\sigma}\right)\log\left(\frac{M|\mathcal{A}|}{\sigma}\right)}\|\boldsymbol{q} - \tilde{\boldsymbol{q}}\|_{\boldsymbol{H}^{-1}}}{\iota^{\frac{1}{6}}}
\end{aligned}
\tag{22}
$$

Here, $\sigma_1, \sigma_2 \in (0, 1)$. Specifically, the second inequality follows from Lemma 2, and the third inequality follows from Lemma 3. By setting $\sigma_1 = \sigma_2 = \frac{\sigma}{2}$ and applying a union bound, the result holds after absorbing constant factors into $C_2$, $C_3$, and $C_4$.

$\square$

## D   Lemmas

**Lemma 1.** *Adapted from Lemma 5.1 of Yang et al. (2020), the regret in Equation 13 can be decomposed as Equation 23, where $\langle\cdot,\cdot\rangle$ means the inner product of two vectors.*

$$
\begin{aligned}
\text{R}_M &= \sum_{m=1}^{M} Q_1^*(s_1^m, u_1^m) - Q_1^{\pi_m}(s_1^m, a_1^m) \\
&= \sum_{m=1}^{M} V_1^*(s_1^m) - V_1^{\pi_m}(s_1^m) \\
&= \sum_{m=1}^{M}\sum_{h=1}^{H}\left[\mathbb{E}_{\pi^*}\left[\delta_h^m(s_h, a_h)|s_1 = s_1^m\right] - \delta_h^m(s_h^m, a_h^m)\right] + \sum_{m=1}^{M}\sum_{h=1}^{H}(\zeta_h^m + \varepsilon_h^m) \\
&\quad + \sum_{m=1}^{M}\sum_{h=1}^{H}\mathbb{E}_{\pi^*}\left[\langle Q_h^m(s_h, \cdot), \pi_h^*(\cdot|s_h) - \pi_m(\cdot|s_h)\rangle |s_1 = s_1^m\right]
\end{aligned}
\tag{23}
$$

**Lemma 2.** *Adapted from Lemma 5.3 of Yang et al. (2020), with probability at least $1 - \sigma_1$, the second term in Equation 22 can be bounded as follows:*

$$
\sum_{m=1}^{M}\sum_{h=1}^{H}(\zeta_h^m + \varepsilon_h^m) \leq \sqrt{16MH^3\log\frac{2}{\sigma_1}}
\tag{24}
$$

**Lemma 3.** *For any $\sigma_2 \in (0, 1)$, assume the width of the action-value network satisfies:*

$$
\iota = \text{poly}(L, D, \frac{1}{\sigma_2}, \log\frac{M|\mathcal{A}|}{\sigma_2})
\tag{25}
$$

where $L$ is the number of layers in the action-value network, and $\mathrm{poly}(\cdot)$ means a polynomial function depending on the incorporated variables, and let:

$$\alpha = \sqrt{2(D \cdot \log(1 + \frac{M \cdot \log|\mathcal{A}|}{\lambda}) - \log \sigma_2)} + \sqrt{\lambda} \tag{26}$$

$$\eta \le C_1(\iota \cdot D^2 M^{\frac{11}{2}} L^6 \cdot \log \frac{M|\mathcal{A}|}{\sigma_2})^{-1} \tag{27}$$

then with probability at least $1 - \sigma_2$, the first term in Equation 22 is bounded as:

$$\sum_{m=1}^{M} \sum_{h=1}^{H} \left[ \mathbb{E}_{\pi^*} \left[ \delta_h^m(s_h, a_h)|s_1 = s_1^m \right] - \delta_h^m(s_h^m, a_h^m) \right] \tag{28}$$

$$\le 4H\sqrt{MH \log \frac{2}{\sigma_2}} + C_2 \alpha H \sqrt{MD \cdot \log(1 + \frac{M}{\lambda D})} + \frac{C_3 \cdot HL^3 D^{\frac{5}{2}} M \sqrt{\log \iota \log\left(\frac{1}{\sigma_2}\right) \log\left(\frac{M|\mathcal{A}|}{\sigma_2}\right)} \|q - \tilde{q}\|_{H^{-1}}}{\iota^{\frac{1}{6}}}$$

*Proof.* According to Yang et al. (2020), there is:

$$\sum_{m=1}^{M} \sum_{h=1}^{H} \left[ \mathbb{E}_{\pi^*} \left[ \delta_h^m(s_h, a_h)|s_1 = s_1^m \right] - \delta_h^m(s_h^m, a_h^m) \right] \le \sum_{m=1}^{M} \sum_{h=1}^{H} -\delta_h^m(s_h^m, a_h^m) \tag{29}$$

Considering $\delta_h^m(s_h^m, a_h^m)$, it can be decomposed as:

$$
\begin{aligned}
\delta_h^m(s_h^m, a_h^m) =& r_h^m + (\mathbb{P}_h V_{h+1}^m)(s_h^m, a_h^m) - Q_h^m(s_h^m, a_h^m) \\
=& r_h^m + (\mathbb{P}_h V_{h+1}^m)(s_h^m, a_h^m) - Q_h^*(s_h^m, a_h^m) + Q_h^*(s_h^m, a_h^m) - Q_h^m(s_h^m, a_h^m) \\
=& \mathbb{P}_h (V_{h+1}^m - V_{h+1}^*)(s_h^m, a_h^m) + (Q_h^* - Q_h^m)(s_h^m, a_h^m) \\
=& \underbrace{\mathbb{P}_h (V_{h+1}^m - V_{h+1}^*)(s_h^m, a_h^m) - (V_{h+1}^m - V_{h+1}^*)(s_{h+1}^m)}_{\omega_h^m} \\
& + \underbrace{(V_{h+1}^m - V_{h+1}^*)(s_{h+1}^m)}_{\rho_{h+1}^m} + \underbrace{(Q_h^* - Q_h^m)(s_h^m, a_h^m)}_{\varphi_h^m}
\end{aligned} \tag{30}
$$

By Azuma-Hoeffding inequality, we can bound $\sum_{m=1}^{M} \sum_{h=1}^{H} \omega_h^m$ as Equation 31 with probability at least $1 - \sigma_3$ where $\sigma_3 \in (0, 1)$.

$$-2H\sqrt{MH \log \frac{2}{\sigma_3}} \le \sum_{m=1}^{M} \sum_{h=1}^{H} \omega_h^m \le 2H\sqrt{MH \log \frac{2}{\sigma_3}} \tag{31}$$

As $\rho_{h+1}^m$ can be decomposed as Equation 32 where $u_{h+1}^m \sim \pi_{h+1}^*(\cdot|s_{h+1}^m)$, there is Equation 33.

$$\rho_{h+1}^m = (V_{h+1}^m - V_{h+1}^*)(s_{h+1}^m) = Q_{h+1}^m(s_{h+1}^m, a_{h+1}^m) - Q_{h+1}^*(s_{h+1}^m, u_{h+1}^m) \tag{32}$$

$$\Rightarrow \sum_{m=1}^M \sum_{h=1}^H (\rho_{h+1}^m + \varphi_h^m) \tag{33}$$

$$= \sum_{m=1}^M \sum_{h=1}^{H-1} \left[ Q_{h+1}^m(s_{h+1}^m, a_{h+1}^m) - Q_{h+1}^*(s_{h+1}^m, u_{h+1}^m) \right] + \sum_{m=1}^M \sum_{h=1}^H (Q_h^* - Q_h^m)(s_h^m, a_h^m)$$

$$= \underbrace{\sum_{m=1}^M \sum_{h=2}^H Q_h^*(s_h^m, a_h^m) - Q_h^*(s_h^m, u_h^m)}_{\mathrm{R_{MAB}}} + \sum_{m=1}^M (Q_1^* - Q_1^m)(s_1^m, a_1^m) \tag{34}$$

Specifically, the second equation follows from $Q_{H+1}^*(s_{H+1}^m, a_{H+1}^m) = 0$ and $Q_{H+1}^m(s_{H+1}^m, a_{H+1}^m) = 0$. The second term in Equation 34 originates from the estimation error of the action-value function, which is constrained by the convergence properties of DQN. Consequently, to complete the proof of Lemma 3, it suffices to establish a bound for the $\mathrm{R_{MAB}}$ term, while the second term is omitted for conciseness in the remaining discussion. Bounds of $\mathrm{R_{MAB}}$ under UCB- and Thompson Sampling-based exploration strategies are proved in Lemma 4 and Lemma 5, respectively.

By setting $C_2 = \max\{C_{\mathrm{ucb}}, C_{\mathrm{ts}}\}$, $\sigma_3 = \sigma_4 = \frac{\sigma_2}{2}$, applying a union bound, and absorbing constant factors into $C_2$ and $C_3$, the proof is completed.

$\square$

### D.1 Regret Bound of UCB-based Exploration

In this subsection, we first introduce the third assumption from *deep representation and shallow exploration* (Xu et al., 2022), which, with Assumptions 1 and 2, forms the standard assumptions. We then present Lemma 4.

**Assumption 3.** *For $\forall s_1, s_2 \in \mathcal{S}$ and $\forall a_1, a_2 \in \mathcal{A}$, there is a constant $l_{Lip} > 0$, such that:*

$$\left\| \nabla_{\boldsymbol{W}} \phi(s_1, a_1 | \boldsymbol{W}_h^1) - \nabla_{\boldsymbol{W}} \phi(s_2, a_2 | \boldsymbol{W}_h^1) \right\|_2 \leq l_{Lip} \left\| (s_1; a_1) - (s_2; a_2) \right\|_2 \tag{35}$$

**Lemma 4.** *Adapted from Theorem 4.4 of Xu et al. (2022), suppose the standard initializations and assumptions hold. Additionally, assume without loss of generality that $\|\boldsymbol{\theta}^*\|_2 \leq 1$ and $\|\phi(s_h, a_h)\|_2 \leq 1$. If with the UCB-based exploration strategy, then for any $\sigma_4 \in (0, 1)$, let the width of the action-value network satisfies:*

$$\iota = \mathrm{poly}(L, D, \frac{1}{\sigma_4}, \log \frac{M|\mathcal{A}|}{\sigma_4}) \tag{36}$$

*where $L$ is the number of layers in the action-value network, and $\mathrm{poly}(\cdot)$ means a polynomial function depending on the incorporated variables, and let:*

$$\alpha = \sqrt{2(D \cdot \log(1 + \frac{M \cdot \log|\mathcal{A}|}{\lambda}) - \log \sigma_4)} + \sqrt{\lambda} \tag{37}$$

$$\eta \leq C_1 (\iota \cdot D^2 M^{\frac{11}{2}} L^6 \cdot \log \frac{M|\mathcal{A}|}{\sigma_4})^{-1} \tag{38}$$

*then with probability at least $1 - \sigma_4$, the term $\mathrm{R}_{\mathrm{UCB}}$ in Equation 34 can be bounded as follows:*

$$\mathrm{R}_{\mathrm{UCB}} \leq C_{\mathrm{ucb}} \cdot \alpha H \sqrt{MD \cdot \log(1 + \frac{M}{\lambda D})} + \frac{C_3 H L^3 D^{\frac{5}{2}} M \sqrt{\log \iota \log \left( \frac{1}{\sigma_4} \right) \log \left( \frac{M|\mathcal{A}|}{\sigma_4} \right)} \| \boldsymbol{q} - \tilde{\boldsymbol{q}} \|_{\boldsymbol{H}^{-1}}}{\iota^{\frac{1}{6}}} \tag{39}$$

*where $C_1, C_{\mathrm{ucb}}, C_3$ are constants independent of the problem parameters; $\boldsymbol{q} = (q_1^1; q_2^1; \ldots; q_H^1; \ldots; q_H^M)$ and $\tilde{\boldsymbol{q}} = (Q_1^1(s_1^1, a_1^1); \ldots; Q_H^1(s_H^1, a_H^1); \ldots; Q_H^M(s_H^M, a_H^M))$ are the target and estimated value vectors, respectively.*

Notably, the proof of the above lemma relies on the concentration of self-normalized stochastic processes. However, since $Q_h^m$ is not independent of historical data, this result cannot be directly applied. Instead, a similar approach to that in Yang et al. (2020) is adopted by leveraging Uniform Convergence across all possible inputs within the value function class $\mathcal{Q}$. This ensures that the maximum deviation between the true and learned values over all time steps remains small, *i.e.*, $\sup_{Q_h^m \in \mathcal{Q}} |Q_h^m - Q_h^*| \leq \epsilon_m$ for $\forall h \in [H]$. Crucially, the error $\epsilon_m$ decreases as the number of episodes increases. Applying triangle inequality, we decompose the error bound $\mathrm{R}_{\mathrm{UCB}}$ into two terms to manage the dependency:

- A true optimal value function component, to which the concentration of self-normalized stochastic processes applies.

- A cumulative error term dependent on $\epsilon_m$, which can be systematically bounded.

## D.2 Regret Bound of Thompson Sampling-based Exploration

**Lemma 5.** *Under the same settings as those of Lemma 4, if with the Thompson Sampling-based exploration strategy, the term $\mathrm{R}_{\mathrm{Thompson\ Sampling}}$ in Equation 34 can be bounded as Equation 40, where $C_{\mathrm{ts}}$ is a constant.*

$$\mathrm{R}_{\mathrm{Thompson\ Sampling}} \leq C_{\mathrm{ts}} \cdot \alpha H \sqrt{MD \cdot \log(1 + \frac{M}{\lambda D})} + \frac{C_3 H L^3 D^{\frac{5}{2}} M \sqrt{\log \iota \log \left( \frac{1}{\sigma_4} \right) \log \left( \frac{M|\mathcal{A}|}{\sigma_4} \right)} \| \boldsymbol{q} - \tilde{\boldsymbol{q}} \|_{\boldsymbol{H}^{-1}}}{\iota^{\frac{1}{6}}} \tag{40}$$

*Proof.* According to Lemma A.1 of Xu et al. (2022), there exits $\boldsymbol{W}_h^{\#}$ such that $Q_h^*(s, u) - Q_h^*(s, a)$ can be decomposed as Equation 41, where $g(s, a; \boldsymbol{W}) = \nabla_{\boldsymbol{W}} \phi(s, a; \boldsymbol{W})$.

$$Q_h^*(s, u) - Q_h^*(s, a) \tag{41}$$

$$= (\boldsymbol{\theta}_h^*)^\mathsf{T} [\phi(s, u; \boldsymbol{W}_h^m) - \phi(s, a; \boldsymbol{W}_h^m)] + (\boldsymbol{\theta}_h^1)^\mathsf{T} \left[ g(s, u; \boldsymbol{W}_h^1) - g(s, a; \boldsymbol{W}_h^1) \right] (\boldsymbol{W}_h^{\#} - \boldsymbol{W}_h^m)$$

$$= (\boldsymbol{\theta}_h^1)^\mathsf{T} \left[ g(s, u; \boldsymbol{W}_h^1) - g(s, a; \boldsymbol{W}_h^1) \right] (\boldsymbol{W}_h^{\#} - \boldsymbol{W}_h^m)$$

$$+ \underbrace{(\boldsymbol{\theta}_h^m)^\mathsf{T} [\phi(s, u; \boldsymbol{W}_h^m) - \phi(s, a; \boldsymbol{W}_h^m)]}_{\vartheta_h^m} - (\boldsymbol{\theta}_h^m - \boldsymbol{\theta}_h^*)^\mathsf{T} [\phi(s, u; \boldsymbol{W}_h^m) - \phi(s, a; \boldsymbol{W}_h^m)]$$

Based on the action selection process using Thompson Sampling-based exploration strategy in Algorithm 3, we derive Equation 42.

$$(\boldsymbol{\theta}_h^m + \alpha \Delta \boldsymbol{\theta}_h^m)^\mathsf{T} \phi(s, u; \boldsymbol{W}_h^m) \leq (\boldsymbol{\theta}_h^m + \alpha \Delta \boldsymbol{\theta}_h^m)^\mathsf{T} \phi(s, a; \boldsymbol{W}_h^m) \tag{42}$$

Consequently, for any $\sigma_5 \in (0, 1)$, with probability at least $1 - \sigma_5$, the term $\vartheta_h^m$ is bounded as Equation 43.

$$\vartheta_h^m \leq \|\Delta\boldsymbol{\theta}_h^m\|_{\boldsymbol{A}_h^m} \|\phi(s, a; \boldsymbol{W}_h^m) - \phi(s, u; \boldsymbol{W}_h^m)\|_{(\boldsymbol{A}_h^m)^{-1}} \tag{43}$$

$$\leq (\sqrt{D} + \sqrt{2\log\frac{1}{\sigma_5}}) \|\phi(s, a; \boldsymbol{W}_h^m) - \phi(s, u; \boldsymbol{W}_h^m)\|_{(\boldsymbol{A}_h^m)^{-1}}$$

Specifically, the last inequality above is because $\Delta\boldsymbol{\theta}_h^m \sim N(0, (\boldsymbol{A}_h^m)^{-1})$. Substituting the bound of $\vartheta_h^m$ back into Equation 41 further yields:

$$Q_h^*(s, u) - Q_h^*(s, a) \leq (\boldsymbol{\theta}_h^1)^\mathsf{T} \left[g(s, u; \boldsymbol{W}_h^1) - g(s, a; \boldsymbol{W}_h^1)\right] (\boldsymbol{W}_h^\# - \boldsymbol{W}_h^m) \tag{44}$$

$$+ (\sqrt{D} + \sqrt{2\log\frac{1}{\sigma_5}}) \|\phi(s, a; \boldsymbol{W}_h^m) - \phi(s, u; \boldsymbol{W}_h^m)\|_{(\boldsymbol{A}_h^m)^{-1}}$$

$$- (\boldsymbol{\theta}_h^m - \boldsymbol{\theta}_h^*)^\mathsf{T} \left[\phi(s, u; \boldsymbol{W}_h^m) - \phi(s, a; \boldsymbol{W}_h^m)\right]$$

Comparing Equation 44 with Equation A.7 of Xu et al. (2022), the difference between the regrets of the Thompson Sampling-based and UCB-based exploration strategies is bounded as in Equation 45.

$$|\mathrm{R}_{\text{Thompson Sampling}} - \mathrm{R}_{\text{UCB}}|$$

$$\leq \sum_{m=1}^{M}\sum_{h=1}^{H}(\sqrt{D} + \sqrt{2\log\frac{1}{\sigma_5}}) \|\phi(s_h^m, a_h^m; \boldsymbol{W}_h^m) - \phi(s_h^m, u_h^m; \boldsymbol{W}_h^m)\|_{(\boldsymbol{A}_h^m)^{-1}}$$

$$+ \sum_{m=1}^{M}\sum_{h=1}^{H}\alpha \|\phi(s_h^m, a_h^m; \boldsymbol{W}_h^m)\|_{(\boldsymbol{A}_h^m)^{-1}} + \sum_{m=1}^{M}\sum_{h=1}^{H}\alpha \|\phi(s_h^m, u_h^m; \boldsymbol{W}_h^m)\|_{(\boldsymbol{A}_h^m)^{-1}}$$

$$\leq C_5\alpha H\sqrt{MD\cdot\log(1+\frac{M}{\lambda D})} \tag{45}$$

Setting $C_{\text{ts}} = C_{\text{ucb}} + C_5$ completes the proof. $\qquad\square$

Notably, the above proof applies a union bound, which requires assigning a constant to $\alpha$ in Equation 37. For brevity, we omit the allocation of this constant and the corresponding discussion of the failure probabilities.

### D.3 Discussion about the Standard Assumptions

Assumption 1 can be readily satisfied by transforming the state–action pairs $(s; a)$ into $(s; a; s; a)$ and applying an appropriate scaling.

Assumption 2 is a mild condition. Specifically, prior studies have demonstrated that for two-layer ReLU networks, this assumption can be directly derived from Asumption 1. A diverse input distribution and a wide neural network can largely ensure that $\boldsymbol{H}$ remains positive definite.

Assumption 3 holds when the gradient of the neural network with respect to certain weights does not fluctuate excessively. In other words, within a neighborhood of a given state–action pair, the gradient with respect to these weights remains well-controlled. This is a standard assumption in the non-convex optimization literature, commonly used to ensure the convergence of alternating optimization procedures in which parameters are updated iteratively.

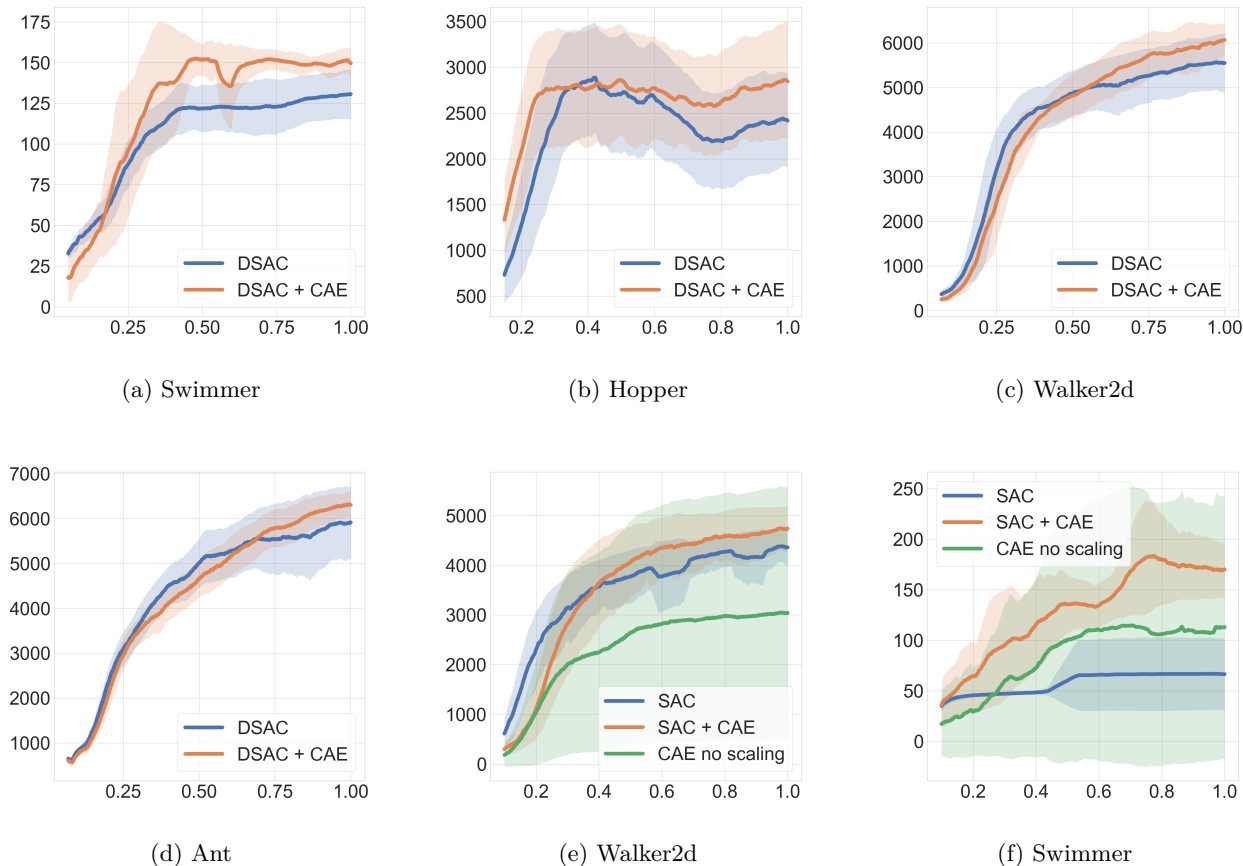

Figure 4: Experimental results of DSAC on MuJoCo-v3, *i.e.*, from Figure 4a to Figure 4d, and those of ablation study to the *scaling strategy* on MuJoCo-v4, *i.e.*, Figure 4e and Figure 4f.

# E  Experiment

In this section, we present additional experimental results and provide the hyperparameters used to reproduce the experiments reported in this paper.

## E.1  Experiment on Mujoco

**Ablation study to the *scaling strategy*.** We conduct experiments to assess the necessity of the scaling strategy. Experiment results, illustrated in Figure 4e and Figure 4f, are based on randomly selected tasks for SAC. As shown, SAC enhanced with CAE fails to deliver satisfactory performance on *Walker2d* and *Swimmer* tasks when the scaling strategy is not applied. It achieves high performance under certain seeds, while it performs poorly under others, leading to poor overall results and large variance. This highlights the critical role of the scaling strategy in ensuring the practical stability and effectiveness of CAE.

Hyperparameters of various RL algorithms for the experiments on MuJoCo are completely the same as those in the public codebase CleanRL. CAE introduces only two more hyperparameters, *i.e.*, the exploration coefficient $\alpha$ and the ridge which is set as $\lambda = 1$. Exploration coefficients are summarized in Table 7 for various tasks and algorithms, and the experimental results on various MuJoCo tasks involving different RL algorithms are in Figure 4 and Figure 5. Notably, we only present results for cases where the *RPI* exceeds 5.0%, as smaller *RPI* is not clearly distinguishable in the figures.

Table 7: Exploration coefficient for various MuJoCo tasks and algorithms

| Algorithms / Env | SAC | PPO | TD3 | DSAC |
|---|---|---|---|---|
| Swimmer | 0.2 | 0.1 | 0.1 | 0.1 |
| Ant | 0.7 | 0.2 | 0.3 | 1.0 |
| Walker2d | 1.0 | 0.13 | 0.8 | 3.0 |
| Hopper | 0.4 | 0.14 | 0.3 | 2.0 |
| HalfCheetah | 0.4 | 0.5 | 3.7 | 3.0 |
| Humanoid | 4.0 | 0.1 | 6.0 | 4.5 |

### E.2 Experiment on MiniHack

In Subsection 5.2, we evaluate our methods CAE and CAE+ on nine representative MiniHack tasks, whose detailed descriptions are provided in Table 8.

Table 8: MiniHack tasks evaluated in Subsection 5.2.

| Short names in Subsection 5.2 | Full names | Task types |
|---|---|---|
| N4 | MultiRoom-N4 | Navigation-based |
| N4-Locked | MultiRoom-N4-Locked | |
| N6 | MultiRoom-N6 | |
| N6-Locked | MultiRoom-N6-Locked | |
| N10-OD | MultiRoom-N10-OpenDoor | |
| Horn | Freeze-Horn-Restricted | Skill-based |
| Random | Freeze-Random-Restricted | |
| Wand | Freeze-Wand-Restricted | |
| LavaCross | LavaCross-Restricted | |

The hyperparameters for IMPALA, E3B, CAE, and CAE+ used in our experiments are summarized in Table 9 and Table 10, aligning with those from the E3B experiments (Henaff et al., 2022). The experimental results on MiniHack are presented in Figure 6.

Table 9: IMPALA Hyperparameters for MiniHack (Henaff et al., 2022)

| | |
|---|---|
| Learning rate | 0.0001 |
| RMSProp smoothing constant | 0.99 |
| RMSProp momentum | 0 |
| RMSProp | $10^{-5}$ |
| Unroll length | 80 |
| Number of buffers | 80 |
| Number of learner threads | 4 |
| Number of actor threads | 8 |
| Max gradient norm | 40 |
| Entropy cost | 0.0005 |
| Baseline cost | 0.5 |
| Discounting factor | 0.99 |

Table 10: E3B, CAE, and CAE+ Hyperparameters for MiniHack

| | | |
|---|---|---|
| E3B, CAE, and CAE+ | Scaling strategy | True |
| | Ridge regularizer | 0.1 |
| | Entropy Cost | 0.005 |
| | Exploration coefficient | 1 |
| CAE+ | Dimension of $U$ | 128 or 256 |

### E.3  Experiment on Habitat

The hyperparameters for PPO, E3B, CAE, and CAE+ used in these experiments are summarized in Table 11 and Table 12.

Table 11: PPO Hyperparameters for Habitat are adopted from *habitat-lab* (Savva et al., 2019; Andrew et al., 2021; Xavi et al., 2023)

| | |
|---|---|
| Clipping | 0.2 |
| PPO epoch | 4 |
| Value loss coefficient | 0.5 |
| Entropy coefficient | 0.01 |
| Learning rate | $2.5e-4$ |
| $\epsilon-$greedy | 1e-5 |
| Max gradient norm | 0.2 |
| Rollout steps | 128 |
| Use GAE | True |
| Discounting factor | 0.99 |
| Number of actor threads | 16 |

Table 12: E3B, CAE, and CAE+ Hyperparameters for Habitat

| | | |
|---|---|---|
| E3B, CAE, and CAE+ | Scaling strategy | False |
| | Ridge regularizer | 0.1 |
| | Inverse Dynamics Model updates per PPO epoch | 3 |
| | Exploration coefficient | 0.1 |
| CAE+ | Dimension of $U$ | 256 |

## F  Limitations

One limitation of the proposed method is that, despite the small number of additional trainable parameters, computing the uncertainty incurs a non-negligible computational overhead due to the need for matrix inversion. This can be mitigated by using approximation techniques, such as the **Rank-1** method described in Subsection 3.3. Additionally, other linear MAB methods could be integrated into the proposed framework to avoid the need to calculate the inverse Gram matrix.

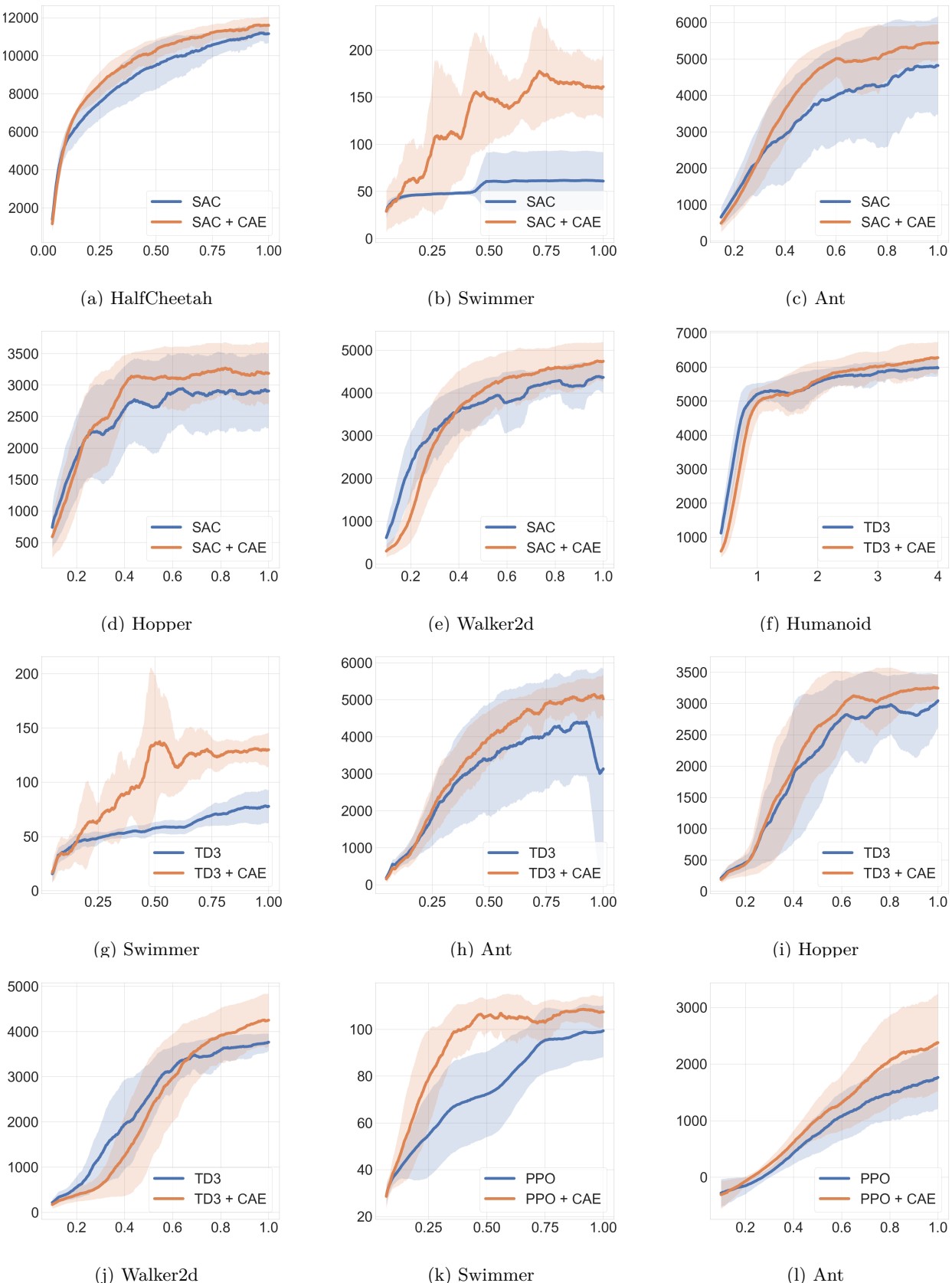

Figure 5: Experimental results on MuJoCo-v4.

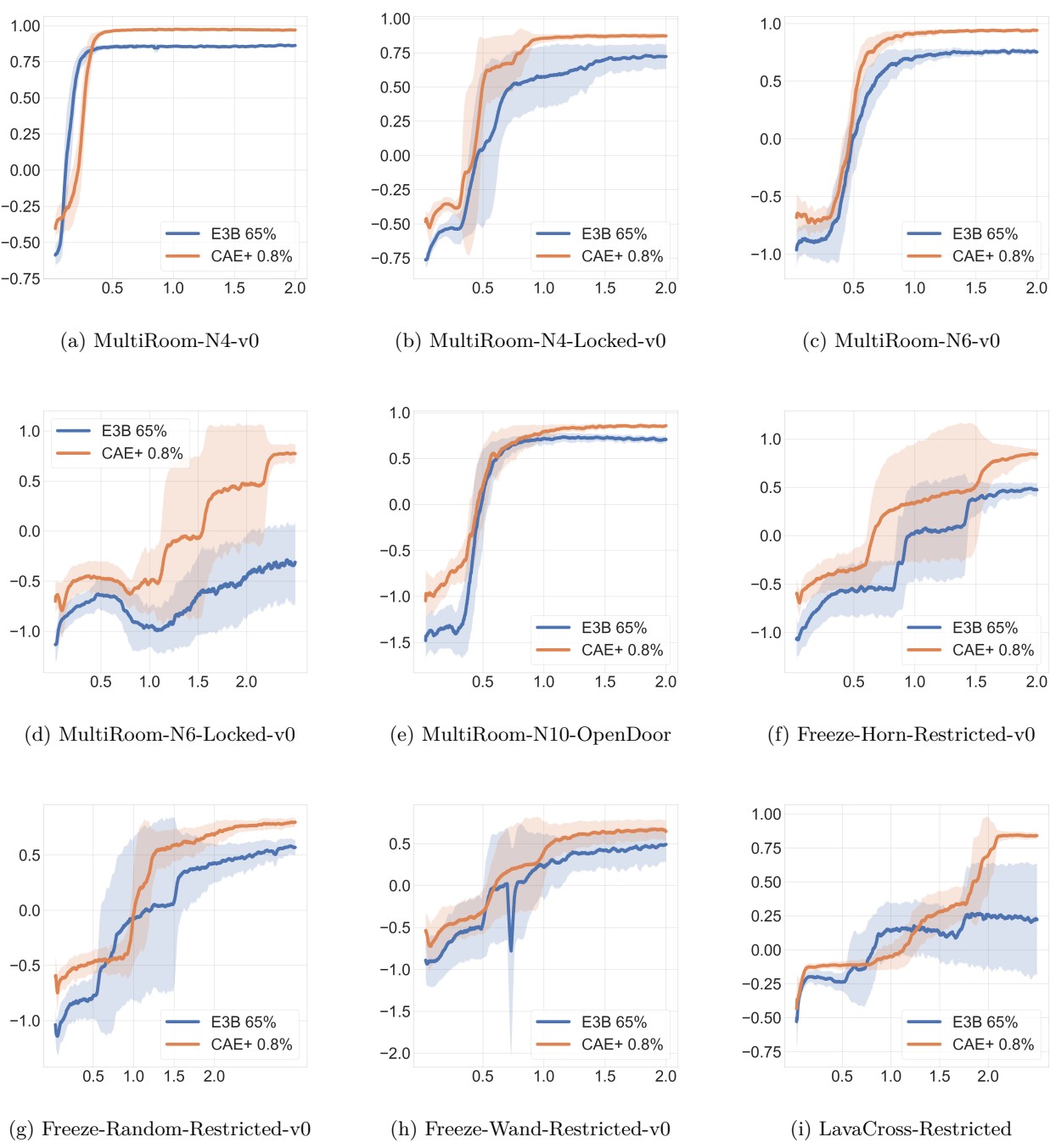

Figure 6: Experimental results on MiniHack.

