# OpenReview forum: "CAE: Repurposing the Critic as an Explorer in Deep Reinforcement Learning"
_TMLR — Accepted by TMLR_

### Review · Reviewer_fQk7 · 2025-11-20

**Summary Of Contributions:**

The paper proposes CAE and CAE+, two lightweight exploration modules that adapt linear MAB techniques to deep RL by leveraging the critic’s learned representation. The methods require minimal additional computation and come with a theoretical regret guarantee. Extensive experiments across MuJoCo, MiniHack, and Habitat, with some competitive baselines, show strong empirical performance.

Strengths:

1. Provides a theoreticalregret bound, with practical applicability.

2. Simple, low-overhead design that integrates easily with existing algorithms.

3. Strong empirical results across diverse environments.

Weaknesses:

1. Conceptual novelty is somewhat limited, as the core idea resembles prior UCB/uncertainty-based exploration methods.

2. Some comparisons (e.g., Table 1) appear overly categorical and would benefit from clearer justification.

**Audience:**

Yes

**Audience Explanation:**

Yes. Exploration in deep reinforcement learning remains a central challenge, and methods with both theoretical guarantees and practical efficiency should be in an interest in the TMLR community.

**Broader Impact Concerns:**

I did not identify any further concern

**Claims And Evidence:**

Yes

**Claims Explanation:**

Overall, most of the main claims are supported by reasonable evidence. The theoretical regret analysis is clearly stated and appears technically sound, and the empirical evaluation is extensive, covering diverse environments with strong baselines. The experimental results consistently support the performance claims. However, some of the comparative statements (eg. Fig 1) are not fully justified and may overstate differences from prior methods. Despite these minor issues, I believe the core claims are generally well supported.

**Requested Changes:**

1. Clarify the criteria used in Table 1.
Although the table provides brief textual definitions of “Linearity,” “Network,” “Deep,” and “Low-overhead,” it is not clear how these criteria were operationalized when assigning ✓/✗ to each method.

For example, NN-UCB employs neural feature extractors, yet is marked as “Deep = ✗”.

And for BDQN (and maybe ACB,  LMCDQN), the “Low-overhead = ✓” entry is questionable. The method relies on ensemble architectures, which will incur additional computational cost. It is unclear why it is classified as low-overhead.

---

> ### Author Response · Authors · 2025-11-26
> **Response to Reviewer fQk7**
>
> Thank you for the valuable time and constructive feedback.
>
> #### **Q1: Clarify the criteria used in Table 1. Although the table provides brief textual definitions of "Linearity", "Network", "Deep" and "Low-overhead", it is not clear how these criteria were operationalized when assigning ✓ or ✗ to each method.**
>
> **A1:**
>
>
>
> We categorize exploration methods into two types:
>
> - **Provable**: methods with rigorous theoretical guarantees.
> - **Empirical**: methods that have been shown to work effectively in practice.
>
> Provable methods can be further subdivided into:
>
> - **Linearity**: methods that assume the underlying functions are linear, which can be overly restrictive.
> - **Network**: methods that use neural networks to approximate value functions.
>
> NN-UCB leverages neural networks while providing theoretical guarantees. Hence, it falls under the **Provable → Network** category and is marked as "✓".
>
> However, due to its high computational cost $O(n^{3})$ per training step, where $n$ is the number of network parameters, NN-UCB cannot be directly applied to **deep RL**, in which $n$ is typically large. This is why it is marked as "✗" under the **Empirical → Deep** category.
>
>
>
> Regarding the **Empirical** category, consider ACB as an example. Although it requires training additional networks, it is computationally feasible. This is why it is marked as "✓" under the **Empirical → Low-overhead** category.
>
> In contrast, LSVI-PHE is marked as "✗" under **Empirical → Low-overhead** because it requires repeatedly sampling independent and identically distributed noise for each **entire trajectory $(m-1)$ times**, i.e., $\sum_{m=1}^{M} (H \times (m-1))$ times in total, where $H$ is the episode length and $M$ is the number of rollouts used for training. Given the typical trajectory lengths and number of rollout episodes in many RL tasks, this process becomes computationally prohibitive in most practical settings.
>
>
>
> We apologize for any confusion. For clarity, we propose updating the terms in the table as follows:
>
> | Provable        |                        | Empirical with Deep RL |                         |
> | --------------- | ---------------------- | ---------------------- | ----------------------- |
> | Linear function | Network-based function | Toy tasks              | Practical complex tasks |
>
>
> We have revised the submission. If the reviewer requires additional details, we would be happy to provide further clarification in the subsequent discussion.
>
> ---
>
>
>
> #### **Weakness 1: Conceptual novelty is somewhat limited, as the core idea resembles prior UCB or uncertainty-based exploration methods.**
>
> **A2:**
>
> We formulate CAE as **a general framework**, rather than a fixed method, allowing various MAB techniques to be adapted to the learned embeddings. Although we focus on two representative algorithms to illustrate the concept and demonstrate effectiveness, the framework itself is **broadly extensible**.
>
> CAE alone may not be sufficient for tasks with complex dynamics or extremely sparse rewards, where learning high-quality value networks is particularly challenging and can hinder exploration that depends on them. To extend the framework to a broader range of practical tasks, we introduce **CAE+**, which incorporates a lightweight inverse dynamics network, making the framework **widely applicable**.
>
> Overall, the proposed framework contributes to the development of sample-efficient and interpretable exploration strategies in deep RL. It not only provides **theoretical support** for exploration but is also **simple to implement** across **a variety of practical scenarios**, ensuring strong **practical performance**. As a result, it has the potential to accelerate progress in applications such as robotics, recommendation systems, autonomous decision-making, and large language models.
>
> We have added this discussion to the **Broader Impact Statement** section.

---

> > ### Comment · Reviewer_fQk7 · 2025-12-20
> > **Thank the authors for the response**
> >
> > Thank the authors for the response. The rebuttal sufficiently address my concerns.

---

> ### Author Response · Authors · 2025-12-31
> **Thank you for the review**
>
> We are very glad to hear that your concerns have been fully addressed. Thank you for your time and thoughtful consideration.

---

### Review · Reviewer_xgnH · 2025-11-20

**Summary Of Contributions:**

This paper proposes Critic as Explorer method, CAE, which repurposes the embedding layers of value networks to compute exploration bonuses using linear MAB techniques. CAE claims to require ~10 lines of code and no extra parameters. For sparse-reward tasks, CAE+ adds a tiny inverse-dynamics network (<1% additional parameters). The method was evaluated on MuJoCo, MiniHack, and Habitat, reporting big improvements over multiple base algorithms and SOTA exploration methods such as E3B.

**Audience:**

Yes

**Audience Explanation:**

The method is easy to implement with minimal additional coding and extra parameters. It potentially has high practical value.

**Claims And Evidence:**

Yes

**Claims Explanation:**

The paper is well-organized with both numerical experiments and theoretical analysis.
The benchmarks are relatively comprehensive.

**Requested Changes:**

1. Add results about the computational cost of CAE.
2. Explicitly discuss the assumptions in the proof, instead of saying “standard assumptions from the literature”.

---

> ### Author Response · Authors · 2025-11-26
> **Response to Reviewer xgnH**
>
> Thank you for the valuable time and constructive feedback.
>
> #### **Q1: Explicitly discuss the assumptions in the proof, instead of saying "standard assumptions from the literature".**
>
> **A1:**
>
> Detailed explanations of the **three assumptions** are provided below, and we have incorporated them into the revised submission.
>
> **Ass. 1.** Assume that $\left\| (s; a) \right\|_{2} = 1$ for $\forall s \in \mathcal{S}, \forall a \in \mathcal{A}$, and that the entries of $(s;a)$ satisfy $(s;a)_j = (s;a)_k$ where $k=j+\frac{D}{2}$ and $D$ represents the dimension of $(s;a)$.
>
> Notably, even if the original state-action pairs do not satisfy this assumption, they can be easily preprocessed to meet it. Specifically, we achieve this by transforming the original state-action pair $(s;a)$ into $(s;a;s;a)$ and apply scaling.
>
> **Ass. 2.** For $\forall s_{1}, s_{2} \in \mathcal{S}$ and $\forall a_{1}, a_{2} \in \mathcal{A}$, there is a constant $l > 0$, such that: $\left\| \nabla_{\boldsymbol W} \phi (s_{1}, a_{1}|\boldsymbol W_{h}^{1}) - \nabla_{\boldsymbol W} \phi (s_{2}, a_{2}|\boldsymbol W_{h}^{1})\right\|_{2} \leq l \Delta$, where $\Delta = \left\| (s_1; a_1) - (s_2; a_2) \right\|_2$.
>
> It is valid under the following conditions:
>
> **1. Gradient Stability**, that the gradient of the neural network with respect to certain weights does not fluctuate excessively. In other words, in the neighborhood of a certain state-action pair, the gradient change remains controlled. This is a standard assumption in **non-convex optimization** research. It ensures the **convergence of alternating optimization** methods, where parameters are updated iteratively.
>
> **2. Input Data Lie in a Benign Subspace**, avoiding extreme cases such as highly nonlinear variations.
>
>
> **Ass. 3.** The neural tangent kernel $\boldsymbol{H}$ of the action-value network is positive definite.
>
> It is a mild condition commonly adopted in related works. Moreover, prior studies have demonstrated that for **two-layer ReLU networks**, this assumption can be directly derived from **Ass .1**. Therefore, **Ass. 3 is mild** in practical RL tasks. A **diverse input distribution** and a **wide neural network** can largely ensure that the **NTK remains positive definite**.
>
>
>
> ---
>
> #### **Q2: Add results about the computational cost of CAE.**
>
> **A2:**
>
> Sorry for this omission. We initially considered the **number of trainable parameters** as a direct reflection of efficiency, given that all other settings— including the base RL algorithm and network architectures—remain identical.
>
> Since RL experiments involve not only training networks but also interacting with **Gym environments on the CPU**, directly analyzing running time is not a common practice in existing works and is not typically considered a standard metric. Therefore, we did not originally include this analysis in our experiments.
>
> However, we did generate a **profile file (.prof)** to analyze script runtimes on MiniHack tasks. Below, we provide an analysis of the **practical running time** based on this profiling data.
>
> | Methods | Total running time | Network storage time | Network serialization time |
> | ------- | ------------------ | -------------------- | -------------------------- |
> | E3B     | $\approx$ 22 hours | 103 seconds          | 48 seconds                 |
> | CAE     | $\approx$ 17 hours | 35.6 seconds         | 28.7 seconds               |
>
> We have incorporated them into the revised submission. Should the reviewer require additional details, we would be happy to provide further clarification in the subsequent discussion.

---

### Review · Reviewer_pnds · 2025-12-05

**Summary Of Contributions:**

In this paper, the authors look to model the uncertainty in estimating the action-value network parameters - the goal is to aid exploration by taking up actions which have not been explored yet until that point. To do so, the authors take inspiration from the literature of MAB and model the uncertainty as done for linear MAB's. The authors provide two algorithms - CAE and CAE+, both of which are efficient and hardly introduce any additional parameters to incorporate the uncertainty/exploration into the model. Extensive experiments are shown with and without CAE for several existing algorithms to validate the updated algorithms.

**Additional Comments:**

Please see my clarification questions in the section for requested changes

**Audience:**

Yes

**Audience Explanation:**

Yes, the uncertainty based updated RL algorithms are important for the community

**Claims And Evidence:**

Yes

**Claims Explanation:**

The claims look good to me - I did not find any glaring issue

**Requested Changes:**

Here are my questions/suggested changes for this paper (the paper has been written very well by the way)

1. It would be good to explain more regarding the uncertainty in linear bandits and why it is an approximation for this case. For instance, the uncertainty is derived theoretically only if W is fixed for calculating the embeddings - that is to capture the uncertainty in $\theta$.

2. Algorithm 1 needs 4 parameters and yet, when it is invoked in Algorithm 2, the parameters are not provided clearly - this led to confusion. Can the authors also provide intuition on why this scaling is necessary in RL algorithms and not in linear bandits?

3. Why is the model for learning the action probabilities (equations 8,9) leading to the uncertainty definition in equation 10? This is a connection I failed to understand. Is the authors somehow saying that the embeddings are projected into a different space via U and hence it is captured in the uncertainty? However, U does not come into the picture for learning the Q function - hence my confusion. Also  Line 14 in Algorithm 2 needs to be expanded (Q function is updated how?) - why does $min(L_f, L_b)$ make sense?

Overall the formulation in CAE+ confused me significantly

4. In the experiments, did the authors compare with random exploration or $epsilon$-greedy or by injecting noise into the actions?

---

> ### Author Response · Authors · 2025-12-09
> **Response to Reviewer pnds**
>
> Thank you for your valuable time and constructive feedback.
>
> #### **Q1. It would be good to explain more regarding the uncertainty in linear bandits and why it is an approximation for this case. For instance, the uncertainty is derived theoretically only if $W$ is fixed for calculating the embeddings - that is to capture the uncertainty in $\theta$.**
>
> **A1:**
>
> Since MAB techniques are applied to the linear layer on top of the embedding layers, the cumulative regret naturally consists of two components, i.e., **the exploration regret from the linear layer**, and **the error induced by the embedding layers' estimation**, the latter appearing as the final term in the regret bound of Theorem 1.
>
> In this final term, as the network estimation error $\left| \boldsymbol{q} - \tilde{\boldsymbol{q}} \right|_{\boldsymbol{H}^{-1}}$ decreases over time, this term typically becomes negligible, consistent with findings in related literature. Therefore, our analysis primarily focuses on the cumulative regret contributed by the linear bandit component, which remains sub-linear in the number of episodes.
>
> We have updated the **Related Work** section to provide additional background on linear bandits and revised the **discussion following Theorem 1** to clarify this approximation in the updated submission.
>
> #### **Q2. Algorithm 1 needs 4 parameters and yet, when it is invoked in Algorithm 2, the parameters are not provided clearly - this led to confusion. Can the authors also provide intuition on why this scaling is necessary in RL algorithms and not in linear bandits?**
>
> **A2:**
>
> The four input parameters of Algorithm 1 are:
>
> - the real-time generated bonus $\phi(s_h^m, a_h^m)$
> - the running mean $\mu$
> - the running variance $\nu^{2}$
> - the running count $\mathcal{N}$
>
> with the last three initialized to $0$.
>
> At each step of the episode, the bonus term $\phi(s_h^m, a_h^m)$ is computed according to Eq. 10, while the running statistics $(\mu, \nu^{2}, \mathcal{N})$ are those from the previous step. These four quantities are then provided to Algorithm 1, which returns the scaled bonus $\phi_h^m$ used for reward shaping, along with the updated running mean $\mu$, variance $\nu^2$, and count $\mathcal{N}$  to be used in the next call to Algorithm 1.
>
> **Intuition behind the scaling strategy:**
>
> 1. In linear bandits, the setting is typically simple, and input contexts can often be normalized to a bounded range, a standard practice in such scenarios.
> 2. In deep RL, however, the intermediate embeddings serve as contexts, and these embedding layers are continuously updated alongside the value network. As a result, exploration based on the constantly changing embedding layers can become highly unstable. Moreover, the varying scale of these embeddings makes proper normalization difficult. The scaling strategy thus serves as a form of normalization, and our experimental results confirm that omitting it leads to significant instability.
>
> We have updated **Algorithm 2** to explicitly include the four parameters passed to Algorithm 1, and added a discussion **before Algorithm 1** to explain its intuition.
>
>
> #### **Q3: Why is the model for learning the action probabilities leading to the uncertainty definition in equation 10? Are the authors somehow saying that the embeddings are projected into a different space via $U$ and hence it is captured in the uncertainty? However, $U$ does not come into the picture for learning the $Q$ function - hence my confusion. Also Line 14 in Algorithm 2 needs to be expanded - why does $\text{min}(L_f, L_b)$ make sense?**
>
> **A3:**
>
> **We have updated Figure 1 to make the framework of CAE+ easier to understand.**
>
> **How $\boldsymbol U$ helps learn the $Q$ function:** Line 14 in Algorithm 2 is $\text{min} (L_f+L_b)$ instead of $\text{min}(L_f, L_b)$, i.e., minimizing the combined loss of the Bellman loss and the auxiliary network loss. As these networks share the embedding layers, minimizing $L_f$, which involves $\boldsymbol U$, facilitates learning the embedding layers of the $Q$ function, and thereby improves $Q$-function learning. Matrix $\boldsymbol U$ learned through minimizing $L_f$ is then employed for **exploration in CAE+**.
>
> **About the rationale for Eq. 10:** Auxiliary network in Eq. 8 learns $\boldsymbol U$ and the embedding layers $\phi(\cdot, \cdot)$.
>
> - In CAE, the embeddings $\phi(s, a)$ serve as contexts, thus the uncertainty by UCB is computed as $    \beta (s, a) = \sqrt{\phi (s, a)^{\mathsf{T}} \boldsymbol A^{-1} \phi (s, a)}$.
> - In CAE+, we transform the contexts via $\boldsymbol U$ to obtain the transformed contexts $\boldsymbol U \phi(s,a)$. Replacing $\phi(s, a)$ with $\boldsymbol U \phi(s, a)$ in the above equation yields the UCB uncertainty as $\sqrt{\phi^{\mathsf{T}} (s, a) \boldsymbol U^{\mathsf{T}} \boldsymbol A^{-1} \boldsymbol U \phi (s, a)}$.
>
> Regarding Thompson Sampling method, the corresponding equation is obtained similarly.
>
> We have updated the **discussion of Eq. 10**.

---

> > ### Author Response · Authors · 2025-12-09
> > **Response to Reviewer pnds**
> >
> > #### **Q4: In the experiments, did the authors compare with random exploration or $\epsilon$-greedy or by injecting noise into the actions?**
> >
> > **A4:**
> >
> > **YES,** we did. In our experiments, the base RL algorithms include PPO, TD3, SAC, DSAC, and IMPALA—all well-established methods that have demonstrated strong performance across a wide range of RL tasks. Each algorithm employs its own exploration strategy, such as $\epsilon$-greedy action selection, action noise injection, or an entropy term, as briefly discussed in the Introduction.
> >
> > As shown in our experimental results, the performance of these standard exploration strategies is clearly inferior to that of our proposed method.

---

### Decision · Action_Editor_UcQW · 2026-01-21

**Recommendation:** Accept with minor revision

**Additional Comments:**

This paper meets TMLR’s acceptance criteria. It addresses a core problem in deep reinforcement learning with a method that is technically sound, lightweight, and of clear interest to both theoretical and applied audiences. The claims are supported by a formal regret analysis and extensive empirical results across dense, sparse, and reward-free environments, and reviewer feedback converges to accept after rebuttal.

Before final publication, the authors are expected to make minor revisions focused on clarity rather than substance. In particular, the exposition of the theoretical approximation underlying the regret bound and the role of learned representations should be made more intuitive, the description of CAE+ (especially the connection between the auxiliary network and the uncertainty formulation) should be clearer, and the algorithms should be fully self-contained and unambiguous. The criteria used in the method comparison table should also be explained more explicitly, and the discussion of computational overhead should be retained and clearly framed.

These changes are editorial and do not require additional experiments or theory.

**Audience:**

Yes

**Audience Explanation:**

Exploration in deep reinforcement learning is a central and long-standing problem, and this paper directly targets an audience within TMLR interested in methods that bridge theory and practice. The proposed CAE and CAE+ frameworks combine provable regret guarantees with lightweight, easily deployable implementations, and are demonstrated across widely used benchmarks and algorithms.
Researchers working on exploration, representation learning in RL, and theoretically grounded yet practical RL methods would find the findings relevant and informative

**Claims And Evidence:**

Yes

**Claims Explanation:**

The submission’s claims are supported by accurate and convincing evidence. The paper provides a formal theoretical analysis with an explicit sub-linear regret bound under stated assumptions, and the derivations are consistent with prior work in neural bandits and deep RL. Empirically, the claims are validated through extensive experiments across dense-reward (MuJoCo), sparse-reward (MiniHack), and reward-free (Habitat) settings, with consistent improvements over strong baselines and state-of-the-art exploration methods. Ablation studies and efficiency analyses further support the necessity of the design choices and the claim that the method is lightweight. While some aspects of the presentation could be clearer, these do not undermine the validity or strength of the evidence.